# Variational Elliptical Processes

**Maria Bånkestad**                                                    *maria.bankestad@it.uu.se*
*RISE Research Institutes of Sweden and Uppsala University, Sweden*

**Jens Sjölund**                                                       *jens.sjolund@it.uu.se*
*Department of Information Technology, Uppsala University, Sweden*

**Jalil Taghia**                                                       *jalil.taghia@ericsson.com*
*Ericsson Research, Sweden*

**Thomas B. Schön**                                                    *thomas.schon@it.uu.se*
*Department of Information Technology, Uppsala University, Sweden*

**Reviewed on OpenReview:** *https://openreview.net/forum?id=djN3TaqbdA&*

## Abstract

We present elliptical processes—a family of non-parametric probabilistic models that subsumes Gaussian processes and Student's $t$ processes. This generalization includes a range of new heavy-tailed behaviors while retaining computational tractability. Elliptical processes are based on a representation of elliptical distributions as a continuous mixture of Gaussian distributions. We parameterize this mixture distribution as a spline normalizing flow, which we train using variational inference. The proposed form of the variational posterior enables a sparse variational elliptical process applicable to large-scale problems. We highlight advantages compared to Gaussian processes through regression and classification experiments. Elliptical processes can supersede Gaussian processes in several settings, including cases where the likelihood is non-Gaussian or when accurate tail modeling is essential.

## 1 Introduction

Systems for autonomous decision-making are increasingly dependent on predictive models. To ensure safety and reliability, it is essential that these models capture uncertainties and risks accurately. Gaussian processes ($\mathcal{GP}$s) offer a framework for probabilistic modeling that is widely used partly because it provides uncertainty estimates. However, these estimates are only reliable to the extent that the model is correctly specified, i.e., that the assumptions of Gaussianity hold true. On the contrary, heavy-tailed data arise in many real-world applications, including finance (Mandelbrot, 1963), signal processing (Zoubir et al., 2012), and geostatistics (Diggle et al., 1998). We use a combination of normalizing flows and modern variational inference techniques to extend the modeling capabilities of $\mathcal{GP}$s to the more general class of elliptical processes ($\mathcal{EP}$s).

**Elliptical processes.** Elliptical processes subsume Gaussian processes and Student's $t$ processes (Rasmussen & Williams, 2006; Shah et al., 2014). They are based on

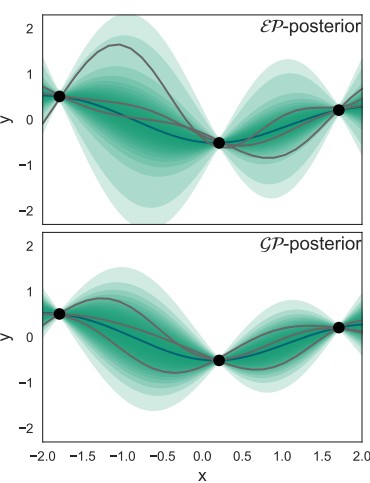

Figure 1: Posterior distributions of elliptical and Gaussian processes with equal kernel hyperparameters and covariance. The shaded areas are credible intervals of the posterior processes. The elliptical credible intervals are wider due to the process' heavier tail.

the elliptical distribution—a scale-mixture of Gaussian distributions that is attractive mainly because it can describe heavy-tailed distributions while retaining most of the Gaussian distribution's computational tractability (Fang et al., 1990). We use a normalizing flow (Rezende & Mohamed, 2015; Papamakarios et al., 2021) to model the continuous scale-mixture, which provides an added flexibility that can benefit a range of applications. We explore using elliptical processes as both a prior (over functions) and a likelihood, as well as the combination thereof. We also explore using $\mathcal{EP}$s as a variational posterior that can adapt its shape to match complex posterior distributions.

**Variational inference.** Variational inference is a tool for approximate inference that uses optimization to find a member of a predefined family of distributions that is close to the target distribution (Wainwright et al., 2008; Blei et al., 2017). Significant advances in the last decade have made variational inference the method of choice for scalable approximate inference in complex parametric models (Ranganath et al., 2014; Hoffman et al., 2013; Kingma & Welling, 2014; Rezende et al., 2014).

It is thus not surprising that the quest for more expressive and scalable variations of Gaussian processes has gone hand-in-hand with the developments in variational inference. For instance, sparse $\mathcal{GP}$s use variational inference to select inducing points to approximate the prior (Titsias, 2009). Inducing points are a common building block in deep probabilistic models such as deep Gaussian processes (Damianou & Lawrence, 2013; Salimbeni et al., 2019) and can also be applied in Bayesian neural networks (Maroñas et al., 2021; Ober & Aitchison, 2021). Similarly, the combination of inducing points and variational inference enables scalable approximate inference in models with non-Gaussian likelihoods (Hensman et al., 2013), such as when performing $\mathcal{GP}$ classification (Hensman et al., 2015; Wilson et al., 2016b).

However, the closeness of the variational distribution to the target distribution is bounded by the flexibility of the variational distribution. Consequently, the success of deep (neural network) models has inspired various suggestions on flexible yet tractable variational distributions, often based on parameterized transformations of a simple base distribution (Tran et al., 2016). In particular, models using a composition of invertible transformations, known as normalizing flows, have been especially popular (Rezende & Mohamed, 2015; Papamakarios et al., 2021).

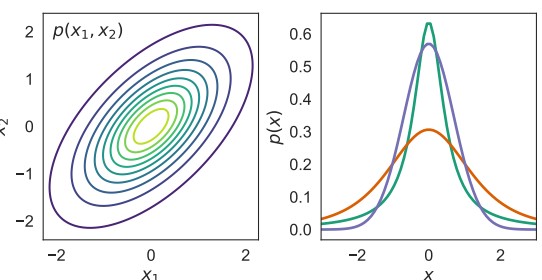

Figure 2: **Left:** A contour plot of an elliptical two-dimensional, correlated distribution with zero means. The name derives from its elliptical level sets. **Right:** Three examples of one-dimensional elliptical distributions with zero means and varying tail-heaviness. Elliptical distributions are symmetric around the mean $\mathbb{E}[\boldsymbol{X}] = \boldsymbol{\mu}$.

**Our contributions.** We propose an adaptation of elliptical distributions and processes in the same spirit as modern Gaussian processes. Our $\mathcal{EP}$ construction is similar to that of Maume-Deschamps et al. (2017) but aims to turn $\mathcal{EP}$s into a practical modeling tool. Constructing elliptical distributions based on a normalizing flow provides a high degree of flexibility without sacrificing computational tractability. This makes it possible to sidestep the "curse of Gaussianity", and adapt to heavy-tailed behavior when called for. We thus foresee many synergies between $\mathcal{EP}$s and recently developed $\mathcal{GP}$ methods. We make a first exploration of these, and simultaneously demonstrate the versatility of the elliptical process as a model for the prior and/or the likelihood, or as the variational posterior. In more detail, our contributions are:

- a construction of the elliptical process and the elliptical likelihood as a continuous scale-mixture of Gaussian processes parameterized by a normalizing flow;
- a variational approximation that can either learn an elliptical likelihood or handle known non-Gaussian likelihoods, such as in classification problems;
- formulating a sparse variational approximation for large-scale problems; and
- describing extensions to heteroscedastic data.

## 2 Background

This section provides the necessary background on elliptical distributions, elliptical processes, and normalizing flow models. Throughout, we consider a regression problem, where we are given a set of $N$ scalar observations, $\boldsymbol{y} = [y_1, \ldots, y_N]^\top$, at the locations $\boldsymbol{X} = [\boldsymbol{x}_1, \ldots, \boldsymbol{x}_N]^\top$, where $\boldsymbol{x}_i$ is $D$-dimensional. The measurement $y_i$ is assumed to be a noisy measurement, such that,

$$y_i = f(\boldsymbol{x}_i) + \epsilon_i, \tag{1}$$

where $\epsilon_i$ is zero mean i.i.d. noise. The task is to infer the underlying function $f : \mathbb{R}^D \to \mathbb{R}$.

### 2.1 Elliptical distributions

Elliptical processes are based on elliptical distributions (Figure 2), which include Gaussian distributions as well as more heavy-tailed distributions, such as the Student's $t$ distribution and the Cauchy distribution.

The probability density of a random variable $Y \in \mathbb{R}^N$ that follows the elliptical distribution can be expressed as

$$p(u; \boldsymbol{\eta}) = c_{N,\boldsymbol{\eta}} |\boldsymbol{\Sigma}|^{-1/2} g_N(u; \boldsymbol{\eta}), \tag{2}$$

where $u := (\boldsymbol{y} - \boldsymbol{\mu})^\top \boldsymbol{\Sigma}^{-1} (\boldsymbol{y} - \boldsymbol{\mu})$ is the squared Mahalanobis distance, $\boldsymbol{\mu}$ is the location vector, $\boldsymbol{\Sigma}$ is the non-negative definite scale matrix, and $c_{N,\boldsymbol{\eta}}$ is a normalization constant. The density generator $g_N(u; \boldsymbol{\eta})$ is a non-negative function with finite integral parameterized by $\boldsymbol{\eta}$, which determines the shape of the distribution.

Elliptical distributions are consistent, i.e., closed under marginalization, if and only if $p(u; \boldsymbol{\eta})$ is a scale-mixture of Gaussian distributions (Kano, 1994). The density can be expressed as

$$p(u; \boldsymbol{\eta}) = |\boldsymbol{\Sigma}|^{-\frac{1}{2}} \int_0^\infty \left(\frac{1}{2\pi\xi}\right)^{\frac{N}{2}} e^{-\frac{u}{2\xi}} \, p(\xi; \boldsymbol{\eta}_\xi) d\xi, \tag{3}$$

using a mixing variable $\xi \sim p(\xi; \boldsymbol{\eta}_\xi)$. Any mixing distribution $p(\xi; \boldsymbol{\eta}_\xi)$ that is strictly positive can be used to define a consistent elliptical process. In particular, we recover the Gaussian distribution if the mixing distribution is a Dirac delta function and the Student's $t$ distribution if it is a scaled inverse chi-square distribution. For more information on the elliptical distribution, see Appendix A.

### 2.2 Elliptical processes

The elliptical process is defined analogously to a Gaussian process as:

**Definition 1** *An elliptical process ($\mathcal{EP}$) is a collection of random variables such that every finite subset has a consistent elliptical distribution, where the scale matrix is given by a covariance kernel.*

This means that an $\mathcal{EP}$ is specified by a mean function $\mu(\boldsymbol{x})$, a scale matrix (a kernel) $k(\boldsymbol{x}, \boldsymbol{x}')$ and the mixing distribution $p(\xi; \boldsymbol{\eta}_\xi)$. Since the $\mathcal{EP}$ is built upon consistent elliptical distributions, it is closed under marginalization. The marginal mean $\boldsymbol{\mu}$ is the same as the mean for the Gaussian distribution, and the covariance is $\mathrm{Cov}[\boldsymbol{Y}] = \mathbb{E}[\xi] \boldsymbol{\Sigma}$ where $\boldsymbol{Y}$ is an elliptical random variable, $\boldsymbol{\Sigma}$ is the covariance for a Gaussian distribution, and $\xi$ is the mixing variable.

Formally, a stochastic process $\{X_t : t \in T\}$ on a probability space $(\Omega, \mathcal{F}, P)$ consists of random maps $X_t : \omega \to S_t, t \in T$, for measurable spaces $(S_t, \mathcal{S}_t), t \in T$ (Bhattacharya & Waymire, 2007). We focus on the setting where $S = \mathbb{R}$ and the index set $T$ is a subset of $\mathbb{R}^N$, in particular, the half-line $[0, \infty)$. Due to Kolmogorov's extension theorem, we may construct the $\mathcal{EP}$ from the family of finite-dimensional, consistent, elliptical distributions, which is due to the restriction to $S = \mathbb{R}$ (which is a Polish space) and Kano's characterization above.

Note that the above definition of the elliptical process is essentially the same as that in Maume-Deschamps et al. (2017) but makes the use of a covariance kernel explicit. (This definition also appears

in Bånkestad et al. (2020), which is an early preprint version of the present article.) Deceptively, there are also definitions of elliptical processes that differ in important ways from ours, notably ones based on Lévy processes (Bingham & Kiesel, 2002) which assume independent increments.

**Identifiability.** When using a $\mathcal{GP}$ for regression or classification, it is typically assumed that the data originate from a single sample path, representing a realization from the $\mathcal{GP}$. However, an elliptical process introduces a hierarchical model wherein $\xi$ is first sampled from the distribution $p(\xi; \boldsymbol{\eta}_\xi)$, followed by sampling $\boldsymbol{f}$ from $\mathcal{GP}(\boldsymbol{f}; \boldsymbol{\mu}, \boldsymbol{K}\xi)$. In this structure, inferring the mixing distribution $p(\xi; \boldsymbol{\eta}_\xi)$ from a single path is impossible since we only have one observation from $p(\xi; \boldsymbol{\eta}_\xi)$. In other words, to infer the mixing distribution $p(\xi; \boldsymbol{\eta}_\xi)$, we must have multiple paths drawn from the same $\mathcal{EP}$. Therefore, to infer an elliptical prior from data, the dataset would have to contain multiple sample paths, such as multiple time series, all originating from the same prior.

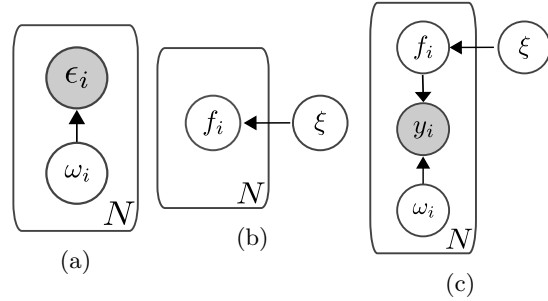

Figure 3: Graphical models of **(a)** the elliptical likelihood, **(b)** the $\mathcal{EP}$-prior, and **(c)** the $\mathcal{EP}$ with independent elliptical noise, where $\omega$ is sampled from the likelihood mixing distribution $p(\omega; \boldsymbol{\eta}_\omega)$.

**Prediction.** To use the $\mathcal{EP}$ for predictions, we need the predictive mean and covariance of the corresponding elliptical distribution. The predictive distribution is guaranteed to be a consistent elliptical distribution but not necessarily the same as the original one—the shape depends on the training samples. (Recall that consistency only concerns the marginal distribution.) The noise-free predictive distribution can be derived analytically (see Appendix B), but to account for additive elliptical noise, we will instead solve it by approximating the posterior $p(\xi | \boldsymbol{y}; \boldsymbol{\eta}_\xi)$ with a variational distribution $q(\xi; \boldsymbol{\varphi}_\xi)$. The approximate inference framework also lets us incorporate (non-Gaussian) noise according to the graphical models in Figure 3.

We aim to model mixing distributions that can capture any shape of the elliptical noise in the data. Thus, to improve upon the piecewise constant parameterization in an earlier version of this work (Bånkestad et al., 2020), we use normalizing flows: a class of methods for learning complex probability distributions.

### 2.3 Flow based models

Normalizing flows are a family of generative models that map simple distributions to complex ones through a series of learned transformations (Papamakarios et al., 2021). Suppose we have a random variable $\boldsymbol{x}$ that follows an unknown probability distribution $p_x(\boldsymbol{x})$. Then, the main idea of a normalizing flow is to express $\boldsymbol{x}$ as a transformation $T_\gamma$ of a variable $\boldsymbol{z}$ with a known simple probability distribution $p_z(\boldsymbol{z})$. The transformation $T_\gamma$ has to be bijective, and it can have learnable parameters $\gamma$. Both $T$ and its inverse have to be differentiable. A change of variables obtains the probability density of $\boldsymbol{x}$:

$$p_x(\boldsymbol{x}) = p_z(\boldsymbol{z}) \left| \det\left( \frac{\partial T_\gamma(\boldsymbol{z})}{\partial \boldsymbol{z}} \right) \right|^{-1}. \tag{4}$$

We focus on one-dimensional flows since we are interested in modeling the mixing distribution. In particular, we use *linear rational spline flows* (Dolatabadi et al., 2020; Durkan et al., 2019), wherein the mapping $T_\gamma$ is an elementwise, monotonic linear rational spline: a piecewise function where each piece is a linear rational function. The parameters are the number of pieces (bins) and the knot locations.

## 3 Method

We propose the variational $\mathcal{EP}$ with elliptical noise, where the variational $\mathcal{EP}$ can learn any consistent elliptical process, and the elliptical noise can capture any consistent elliptical noise. The key idea is to model the

mixing distributions with a normalizing flow. The joint probability distribution of the model (see Figure 3c) is

$$p(\boldsymbol{y}, \boldsymbol{f}, \omega, \xi; \boldsymbol{\eta}) = \underbrace{p(\boldsymbol{f}|\xi; \boldsymbol{\eta_f})p(\xi; \boldsymbol{\eta_\xi})}_{\text{prior}} \underbrace{\prod_{i=1}^{N} p(y_i|f_i, \omega_i)p(\omega_i; \boldsymbol{\eta_\omega})}_{\text{likelihood}}. \tag{5}$$

Here, $p(\boldsymbol{f}|\xi; \boldsymbol{\eta_f}) \sim \mathcal{N}(0, \boldsymbol{K}\xi)$ is a regular $\mathcal{GP}$ prior with the covariance kernel $\boldsymbol{K}$ containing the parameters $\boldsymbol{\eta_f}$, $p(\xi; \boldsymbol{\eta_\xi})$ is the process mixing distribution, and $p(\omega; \boldsymbol{\eta_\omega})$ is the noise mixing distribution.

To learn the mixing distributions $p(\xi; \boldsymbol{\eta_\xi})$ and $p(\omega; \boldsymbol{\eta_\omega})$ by gradient-based optimization, they need to be differentiable with respect to the parameters $\boldsymbol{\eta_\xi}$ and $\boldsymbol{\eta_\omega}$ in addition to being flexible and computationally efficient to evaluate and sample from. Based on these criteria, a spline flow (Section 2.3) is a natural fit. We construct the mixing distributions by transforming a sample from a standard normal distribution with a spline flow. The output of the spline flow is then projected onto the positive real axis using a differentiable function such as *Softplus* or *Sigmoid*.

In the following sections, we detail the construction of the model and show how to train it using variational inference. For clarity, we describe the likelihood first before combining it with the prior and describing a (computationally efficient) sparse approximation.

### 3.1 Likelihood

By definition, the likelihood (Figure 3a) describes the measurement noise $\epsilon_i$ (Equation (1)). The probability distribution of the independent elliptical likelihood is

$$p(\epsilon_i; \sigma, \boldsymbol{\eta_\omega}) = \int \mathcal{N}(\epsilon_i; 0, \sigma^2 \omega_i)p(\omega_i; \boldsymbol{\eta_\omega})d\omega_i, \tag{6}$$

where $\sigma > 0$, and can be set to unity without loss of generality. In other words, the likelihood is a continuous mixture of Gaussian distributions where, e.g., $\epsilon_i$ follows a Student's $t$ distribution if $\omega$ is scaled inverse chi-squared distributed.

**Parameterization.** We parameterize $p(\omega; \boldsymbol{\eta_\omega})$ as a spline flow,

$$p(\omega; \boldsymbol{\eta_\omega}) = p(\zeta)\left|\frac{\partial T(\zeta; \boldsymbol{\eta_\omega})}{\partial \zeta}\right|^{-1}, \tag{7}$$

although it could be, in principle, any positive, finite probability distribution. Here, $p(\zeta) \sim \mathcal{N}(0, 1)$ is the base distribution and $\omega = T(\zeta; \boldsymbol{\eta_\omega})$ represents the spline flow transformation followed by a *Softplus* transformation to guarantee positivity of $\omega$. The flexibility of this flow-based construction lets us capture a broad range of elliptical likelihoods, but we could also specify an appropriate likelihood ourselves. For instance, using a categorical likelihood enables $\mathcal{EP}$ classification; see Section 4.5.

**Training objective.** Now, suppose that we observe $N$ independent and identically distributed residuals $\epsilon_i = y_i - f_i$ between the observations $\boldsymbol{y}$ and some function $\boldsymbol{f}$. We are primarily interested in estimating the noise for the purpose of "denoising" the measurements. Hence, we fit an elliptical distribution to the residuals by maximizing the (log) marginal likelihood with respect to the parameters $\boldsymbol{\eta_\omega}$, that is

$$\log p(\boldsymbol{\epsilon}; \boldsymbol{\eta_\omega}) = \sum_{i=1}^{N} \log \int \mathcal{N}(\epsilon_i; 0, \omega_i) \, p(\omega_i; \boldsymbol{\eta_\omega})d\omega_i. \tag{8}$$

For general mixing distributions, this integral lacks a closed-form expression, but since it is one-dimensional we can approximate it efficiently by numerical integration (for example, using the trapezoidal rule).

Ultimately, we arrive at the likelihood

$$p(\boldsymbol{y}|\boldsymbol{f}) = \prod_{i=1}^{N} \int \mathcal{N}(y_i; f_i, \omega_i) \, p(\omega_i; \boldsymbol{\eta_\omega})d\omega_i. \tag{9}$$

## 3.2 Prior

Recall that our main objective is to infer the latent *function* $f^* = f(\boldsymbol{x}^*)$ at arbitrary locations $\boldsymbol{x}^* \in \mathbb{R}^D$ given a finite set of noisy observations $\boldsymbol{y}$. In probabilistic machine learning, the mapping $\boldsymbol{y} \mapsto f^*$ is often defined by the posterior predictive distribution

$$p(f^*|\boldsymbol{y}) = \int p(f^*|\boldsymbol{f})p(\boldsymbol{f}|\boldsymbol{y})d\boldsymbol{f}, \tag{10}$$

which turns modeling into a search for suitable choices of $p(f^*|\boldsymbol{f})$ and $p(\boldsymbol{f}|\boldsymbol{y})$. Accordingly, the noise estimation described in the previous section is only done in pursuit of this higher purpose.

**Sparse formulation.** For an elliptical process ($\mathcal{EP}$) we can rewrite the posterior predictive distribution as

$$p(f^*|\boldsymbol{y}) = \int p(f^*|\boldsymbol{f},\xi)\,p(\boldsymbol{f},\boldsymbol{u},\xi|\boldsymbol{y})d\boldsymbol{f}\,d\boldsymbol{u}\,d\xi, \tag{11}$$

where we are marginalizing not only over the mixing variable $\xi$ and the function values $\boldsymbol{f}$ (at the given inputs $\boldsymbol{X}$), but also over the function values $\boldsymbol{u}$ at the so-called $M$ inducing inputs $\boldsymbol{Z}$. Introducing inducing points lets us derive a *sparse* variational $\mathcal{EP}$—a computationally scalable version of the $\mathcal{EP}$ similar to the sparse variational $\mathcal{GP}$ (Titsias, 2009).

The need for approximation arises because of the intractable second factor $p(\boldsymbol{f},\boldsymbol{u},\xi|\boldsymbol{y})$ in Equation (11). (The first factor $p(f^*|\boldsymbol{f},\xi)$ is simply a Normal distribution.) We summarize the sparse variational $\mathcal{EP}$ below and refer to Appendices D and E for additional details.

**Variational approximation.** We make the variational ansatz $p(\boldsymbol{f},\boldsymbol{u},\xi|\boldsymbol{y}) \approx p(\boldsymbol{f}|\boldsymbol{u},\xi)q(\boldsymbol{u}|\xi)\,q(\xi)$, where $\boldsymbol{u}$ is the latent function at the induced input locations $\boldsymbol{Z}$, and parameterize this variational posterior as an elliptical distribution. We do so for two reasons: first, this makes the variational posterior similar to the true posterior, and second, we can then use the conditional distribution to make predictions. In full detail, we factorize the posterior as

$$q(\boldsymbol{f},\boldsymbol{u},\xi;\,\boldsymbol{\varphi}) = p(\boldsymbol{f}|\boldsymbol{u},\xi;\,\boldsymbol{\eta_f})q(\boldsymbol{u}|\xi;\,\boldsymbol{\varphi_u})q(\xi;\,\boldsymbol{\varphi_\xi}), \tag{12}$$

where $\boldsymbol{\varphi} = (\boldsymbol{\varphi_f},\boldsymbol{\varphi_u},\boldsymbol{\varphi_\xi})$ are the variational parameters, $q(\boldsymbol{u}|\xi;\,\boldsymbol{\varphi_u}) = \mathcal{N}(\boldsymbol{m},\boldsymbol{S}\xi)$ is a Gaussian distribution with variational parameters $\boldsymbol{m}$ and $\boldsymbol{S}$, and the mixing distribution $\xi \sim q(\xi;\,\boldsymbol{\varphi_\xi})$. Again, $q(\xi;\,\boldsymbol{\varphi_\xi})$ could be any positive finite distribution, but since we only know that the posterior is elliptical with a data-dependent mixing distribution, and since it is impossible to "overfit" a variational distribution (Bishop & Nasrabadi, 2006), we choose a very flexible yet tractable model, namely a spline flow.

Note that, because of the conditioning on $\xi$, the first two factors in Equation (12) are a Gaussian conjugate pair in $\boldsymbol{u}$. Thus, marginalization over $\boldsymbol{u}$ results in a Gaussian distribution, for which the marginals of $f^*$ only depend on the corresponding input $\boldsymbol{x}^*$ (Salimbeni et al., 2019):

$$q(f^*|\xi;\boldsymbol{\varphi}) = \mathcal{N}(f^*|\mu_{\boldsymbol{f}}(\boldsymbol{x}^*),\sigma_{\boldsymbol{f}}(\boldsymbol{x}^*)\xi), \tag{13}$$

where

$$\mu_{\boldsymbol{f}}(\boldsymbol{x}^*) = \boldsymbol{k}_*^\top \boldsymbol{K_{uu}}^{-1}\boldsymbol{m}, \tag{14}$$

$$\sigma_{\boldsymbol{f}}(\boldsymbol{x}^*) = k_{**} - \boldsymbol{k}_*^\top \left(\boldsymbol{K_{uu}}^{-1} - \boldsymbol{K_{uu}}^{-1}\boldsymbol{S}\boldsymbol{K_{uu}}^{-1}\right)\boldsymbol{k}_*, \tag{15}$$

and $k_{**} = k(\boldsymbol{x}^*,\boldsymbol{x}^*)$, $\boldsymbol{k}_* = k(\boldsymbol{Z},\boldsymbol{x}^*)$, and $\boldsymbol{K_{uu}} = k(\boldsymbol{Z},\boldsymbol{Z})$.

Predictions on unseen data points $\boldsymbol{x}^*$ are then computed according to (see Appendix E)

$$q(f^*|\boldsymbol{y};\boldsymbol{x}^*) = \mathbb{E}_{q(\xi;\,\boldsymbol{\varphi_\xi})}\left[\mathcal{N}(f^*|\mu_{\boldsymbol{f}}(\boldsymbol{x}^*),\sigma_{\boldsymbol{f}}(\boldsymbol{x}^*)\xi)\right]. \tag{16}$$

For training, we use variational inference (VI), i.e., maximizing the evidence lower bound (ELBO) to indirectly maximize the marginal likelihood. We train the model using stochastic gradient descent and black-box variational inference (Bingham et al., 2019; Wingate & Weber, 2013; Ranganath et al., 2014).

**VI training.** The marginal likelihood is

$$p(\boldsymbol{y};\,\boldsymbol{\eta_f},\boldsymbol{\eta_u},\boldsymbol{\eta_\xi}) = \int p(\boldsymbol{y},\boldsymbol{f},\boldsymbol{u},\xi;\,\boldsymbol{\eta_f},\boldsymbol{\eta_u},\boldsymbol{\eta_\xi})d\boldsymbol{f}d\boldsymbol{u}d\xi = \int p(\boldsymbol{y}|\boldsymbol{f})p(\boldsymbol{f}|\boldsymbol{u},\xi;\,\boldsymbol{\eta_f})p(\boldsymbol{u},\xi;\,\boldsymbol{\eta_u},\boldsymbol{\eta_\xi})d\boldsymbol{f}d\boldsymbol{u}d\xi. \quad (17)$$

This integral is intractable since $p(\xi;\,\boldsymbol{\eta_\xi})$ is parameterized by a spline flow. To overcome this we approximate the marginal likelihood with the ELBO

$$\mathcal{L}_{\mathrm{ELBO}}(\boldsymbol{\eta_f},\boldsymbol{\eta_u},\boldsymbol{\eta_\xi},\boldsymbol{\varphi_f},\boldsymbol{\varphi_u},\boldsymbol{\varphi_\xi}) = \mathbb{E}_{q(\boldsymbol{f},\xi;\,\boldsymbol{\varphi})}\left[\log p(\boldsymbol{y}|\boldsymbol{f})\right] - D_{\mathrm{KL}}\left(q(\boldsymbol{u},\xi;\,\boldsymbol{\varphi})\,||\,p(\boldsymbol{u},\xi;\,\boldsymbol{\eta})\right)$$
$$= \sum_{i=1}^{N}\mathbb{E}_{q(f_i,\xi;\,\boldsymbol{\varphi})}\left[\log p(y_i|f_i)\right] - D_{\mathrm{KL}}\left(q(\boldsymbol{u},\xi;\,\boldsymbol{\varphi})\,||\,p(\boldsymbol{u},\xi;\,\boldsymbol{\eta})\right). \quad (18)$$

Had the likelihood been Gaussian, the expectation $\mathbb{E}_{q(f_i,\xi;\,\boldsymbol{\varphi})}\left[\log p(y_i|f_i;\,\boldsymbol{\eta_f})\right]$ could have been computed analytically. In our case, however, it is elliptical, and we use a Monte Carlo estimate instead. Inserting the elliptical likelihood (Equation (9)) from the previous section yields

$$\mathcal{L}(\boldsymbol{\eta},\boldsymbol{\varphi}) = \sum_{i=1}^{N}\mathbb{E}_{q(f_i,\xi;\,\boldsymbol{\varphi})}\left[\log\left(\mathcal{N}\left(y_i;f_i,\omega_i\right)p(\omega_i,\boldsymbol{\eta_\omega})\right)\right] - D_{\mathrm{KL}}\left(q(\boldsymbol{u},\xi;\,\boldsymbol{\varphi})||p(\boldsymbol{u},\xi;\,\boldsymbol{\eta})\right). \quad (19)$$

### 3.3 Extension to heteroscedastic noise

We now extend the elliptical likelihood to capture heteroscedastic (input-dependent) noise. The main idea is to let the parameters $\boldsymbol{\eta}_{\omega_i}$ of the likelihood's mixing distribution depend on the input location $\boldsymbol{x}_i$.

In heteroscedastic regression, the noise depends on the input location $\boldsymbol{x}_i$. For example, heteroscedastic elliptical noise can be useful in a time series where the noise variance and tail-heaviness change over time. Examples of this can be found in statistical finance (Liu et al., 2020) and robotics (Kersting et al., 2007). To model this, we use a neural network $g_{\boldsymbol{\gamma}_\omega}(\cdot)$ with parameters $\boldsymbol{\gamma}_\omega$ to represent the mapping from input location to spline flow parameters, $\boldsymbol{x}_i \xmapsto{g_{\boldsymbol{\gamma}_\omega}(\cdot)} \boldsymbol{\eta}_{\omega_i}$.

We train the model by maximizing the log-likelihood

$$\mathcal{L}(\boldsymbol{\gamma}_\omega) = \sum_{i=1}^{N}\log\int p(y_i|f_i,\,\omega_i)p(\omega_i;\,\boldsymbol{\eta}_{\omega_i} = g_{\boldsymbol{\gamma}_\omega}(\boldsymbol{x}_i))d\omega_i. \quad (20)$$

Additional information, such as time of the day or season for time series data, can be incorporated by simply passing it as extra inputs to the neural network $g_{\boldsymbol{\gamma}_\omega}(\cdot)$.

## 4 Experiments

We examined the variational elliptical processes using four different experiments. In the **first** experiment, we investigated how well the elliptical likelihood (Section 3.1) recovers known elliptical noise in synthetic data. In the **second** experiment, we demonstrated the use of the heteroscedastic $\mathcal{EP}$ on a synthetic dataset. We evaluated regression performance on seven standard benchmarks in the **third** experiment where we compared the sparse and the heteroscedastic $\mathcal{EP}$ formulations with both sparse $\mathcal{GP}$ ($\mathcal{SVGP}$) (Hensman et al., 2013) and full $\mathcal{GP}$ baselines. Finally, in the **fourth** experiment, we examined if using an $\mathcal{EP}$ is beneficial in classification tasks.

**Implementation.** The mixing distribution of the variational $\mathcal{EP}$ used a quadratic rational spline flow, where we transformed the likelihood flow $p(\omega)$ using *Softplus* and the posterior flow $p(\xi)$ using *arctan* to ensure that they were positive (remember that a mixing distribution must be positive, see Equation (3)). We used a squared exponential kernel with independent length scales in all experiments. See Appendix F for further implementation details. Accompanying code is found at .

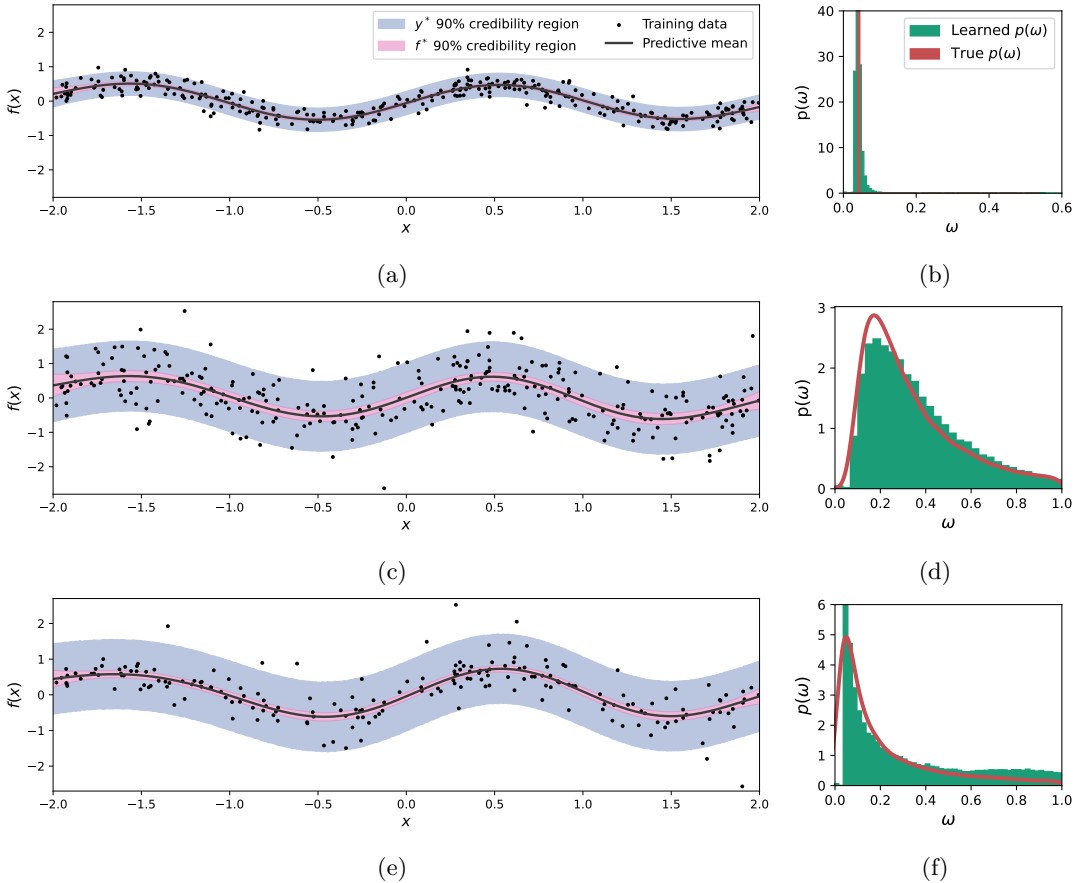

Figure 4: The posterior predictive distribution when using a $\mathcal{GP}$ with elliptical noise modeled by a spline flow. Each row represents a synthetic dataset with different noise. The **first** row adds Gaussian noise $\omega \sim \delta(\omega - 0.04)$ the **second** row adds Student's $t$ noise $\omega \sim$Scale-Inv-$\chi^2(\nu = 4)$, and the **third** row adds Cauchy noise $\omega \sim$Scale-Inv-$\chi^2(\nu = 1)$. The shaded areas show the latent function posterior $f^*$ and the noisy posterior $y^*$ 90% credibility areas. The histograms show the learned and the true noise mixing distribution.

## 4.1 Noise identification

To examine how well the elliptical likelihood, described in Section 3.1, captures different types of elliptical noise, we created three synthetic datasets with the same latent function $f_i = \sin(3x_i)/2$ sampled independently at $N = 200$ locations $x_i \sim U(-2, 2)$. Further, each dataset was contaminated with independent elliptical noise $\epsilon_i$ that was added to the latent function, $y_i = f_i + \epsilon_i$. The added noise varied for the three datasets. The first was Gaussian distributed which is the same as $\omega$ being Delta distributed $\omega \sim \delta(\omega - 0.04)$. The second was Student's $t$ which means that $\omega$ follows the scaled inverse chi-squared distribution $\omega \sim$Scale-Inv-$\chi^2(\nu = 4)$. The third was Cauchy distributed $\omega \sim$Scale-Inv-$\chi^2(\nu = 1)$. We trained a sparse variational $\mathcal{GP}$ for each dataset with an elliptical likelihood.

Figure 4 illustrates the results from the experiments. The histograms in the right column compare the learned mixing distribution $p(\omega; \boldsymbol{\eta}_\omega)$ to the true mixing distribution (the red curve) from which the noise $\epsilon_i$ originated. The learned distributions follow the shape of the true mixing distribution reasonably well, considering the small number of samples, indicating that we can learn the correct likelihood regardless of the noise variance. Furthermore, if the noise is actually Gaussian, as at $x = -0.7$, then so is the learned likelihood. The left column shows the predictive posterior of the final models, demonstrating that the models managed to learn suitable kernel parameters jointly with the likelihood.

**Robust regression on synthetic data.** An elliptical likelihood is better at handling outliers and non-Gaussian noise than a Gaussian likelihood because it can better match the whole distribution of the noise rather than just a single variance. This is shown in Figure 5, where a $\mathcal{GP}$ with an elliptical and a Gaussian likelihood were trained on a small synthetic dataset with additive Student's $t$ noise, with $\eta = 4$. The Gaussian likelihood approximates the mixing distribution with a single variance at approximately $\omega = 0.4$, while the elliptical likelihood fits the entire mixing distribution. As a result, the $\mathcal{GP}$ with the Gaussian likelihood needs to use a shorter length scale to compensate for the thin tail of the likelihood. In contrast, the $\mathcal{GP}$ with the elliptical likelihood can focus on the slower variations, thus producing a better fit to data.

## 4.2 Elliptic heteroskedastic noise

In this experiment, we aimed to exemplify the benefits of using heteroscedastic elliptical noise as described in Section 3.3. To this end, we created the synthetic dataset shown in Figure 6. It consisted of 150 samples generated by adding heteroscedastic noise to the function $f(x_i) = \sin(5x_i) + x_i$, where $x_i \sim U(0, 4)$. Specifically, we added Student's $t$ noise, $\epsilon(x_i) \sim St(\nu(x_i), \sigma(x_i))$, where the noise scale followed $\nu(x_i) = 25 - 11|x_i + 1|^{0.9}$, and the standard deviation by $\sigma(x_i) = 0.5|x_i + 1|^{1.6} + 0.001$. We used a variational sparse $\mathcal{GP}$ with heteroscedastic noise as described in Section 3.3.

The experimental results, depicted in Figure 6, show that, qualitatively, even though the rapid change in the noise

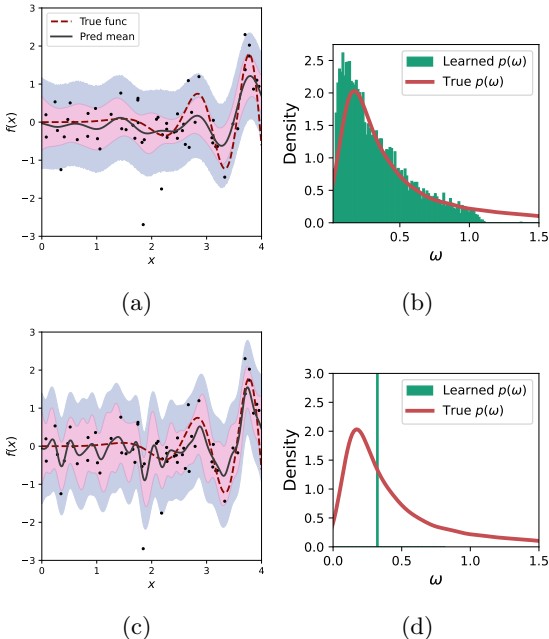

(a)        (b)

(c)        (d)

Figure 5: The predictive posterior after training on a small synthetic dataset. The shaded areas show the 95 % credibility area of the latent function posterior $f^*$ (blue) and the noisy posterior $y^*$ (magenta) when using (**top row**) a $\mathcal{GP}$ with elliptical noise modeled by a spline flow and a (**bottom row**) a $\mathcal{GP}$ with Gaussian noise. The histograms show the learned and the true noise mixing distribution.

distribution and the low number of training samples, the model captures the varying noise in terms of the scale and the increasing heaviness of the tail. Remember that a single spike in the mixing distribution, as at $x = 0.7$, indicates that the noise is Gaussian, and the *wider* the mixing distribution is, as at $x = -0.7$, the heavier-tailed the noise is. When the synthetic data has Gaussian noise, then so has also the learned elliptical noise. Similarly, when the synthetic noise is heavier-tailed, so is the learned mixing distribution. This indicates that this model could be helpful for data with varying elliptical noise.

## 4.3 Regression

We conducted experiments on seven datasets from the UCI repository (Dua & Graff, 2017) to study the impact of the elliptical noise, elliptical posterior, and heteroscedastic noise. We used various regression models based on a Gaussian Process ($\mathcal{GP}$) prior; see Table 1 for a summary. As baselines, we compare with the sparse variational $\mathcal{GP}$ model of Hensman et al. (2013), which we call $\mathcal{SVGP}$, and an exact $\mathcal{GP}$.

**Models evaluated.** We used a $\mathcal{GP}$ model with elliptical noise ($\mathcal{EP}-\mathcal{GP}$) to compare its performance to the traditional $\mathcal{GP}$ model with Gaussian noise. Theoretically, an elliptical posterior should result from combining a Gaussian prior and an elliptical likelihood, but in this case, we approximated the posterior with a Gaussian. We also included a model that used an elliptical posterior ($\mathcal{EP} - \mathcal{EP}$) to explore the potential benefits of using the theoretically more accurate elliptical posterior. Additionally, we tested a heteroscedastic elliptical noise model (Het-$\mathcal{EP}$) and a heteroscedastic Gaussian noise model (Het-$\mathcal{GP}$) to compare their performance. The difference between these two is that in Het-$\mathcal{GP}$ the neural network only predicts the noise variance, whereas the Het-$\mathcal{EP}$ model predicts the 26 parameters corresponding to nine bins of the spline flow.

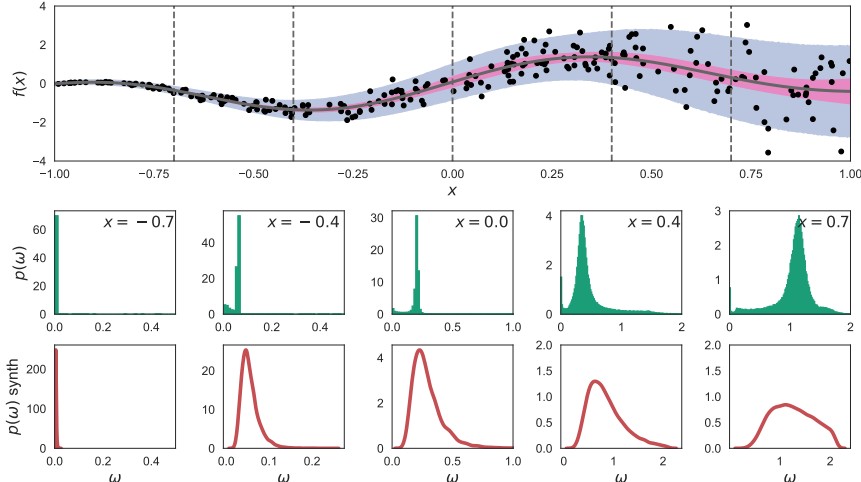

Figure 6: The results from training a $\mathcal{GP}$ with heteroscedastic elliptical noise on a synthetic dataset. The **top** row shows the posterior distribution where the shaded areas are the 95 % credibility areas of the latent posterior $f^*$ (magenta) and the noisy posterior $y^*$ (blue). The histograms in the **middle** row show the noise mixing distributions at the different $x$-values indicated by the vertical dashed lines in the top plot. The **bottom** row shows the mixing distribution used when creating the synthetic data.

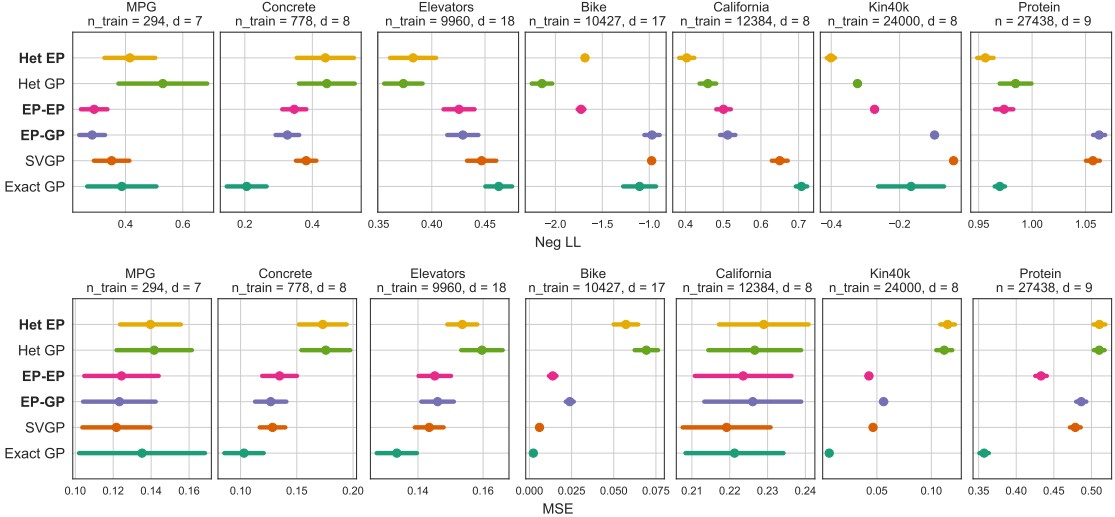

Figure 7: Predictive negative log-likelihood (LL) (**top**) and mean squared error (MSE) (**bottom**) on held-out test data from the regression benchmarks (smaller is better). We show the average of the ten splits as a dot and the standard deviation as a line. The models with bold fonts are our models. Note that the spread of the error varies between the datasets. For example, the MSE error for the Bike dataset is low for all six models. Overall, the $\mathcal{EP}$ posterior outperforms the $\mathcal{GP}$ posterior, regarding the log-likelihood, for the five larger datasets .

First, we summarize the results in Figure 7, and then we discuss the results of each method in more detail. The figure displays the mean and standard deviation of ten randomly chosen training, validation, and test data splits. The training procedure for all models optimizes —directly or indirectly—the log-likelihood. Therefore, the most relevant figure of merit is the negative test log-likelihood (LL), shown in the top row of Figure 7. We stress that log-likelihood is more critical than MSE because mean squared error (MSE) does not consider the predictive variance. However, we show the MSE on held-out test sets in the bottom row for completeness. Additional details on the experiments can be found in Appendix G.

Table 1: The different types of models we trained on the regression datasets.

| NAME | APPROXIMATION | LOSS | LIKELIHOOD | POSTERIOR |
|------|---------------|------|------------|-----------|
| Exact $\mathcal{GP}$ | Exact | Marginal likelihood | Gaussian | Gaussian |
| $\mathcal{SVGP}$ | Variational | ELBO | Gaussian | Gaussian |
| $\mathcal{EP}$-$\mathcal{GP}$ | Variational | ELBO | Elliptic | Gaussian |
| $\mathcal{EP}$-$\mathcal{EP}$ | Variational | ELBO | Elliptic | Elliptic |
| Het-$\mathcal{GP}$ | Variational | ELBO | Gaussian | Gaussian |
| Het-$\mathcal{EP}$ | Variational | ELBO | Elliptic | Gaussian |

$\mathcal{GP}$ **baseline.** To assess the quality of the approximations introduced, we first establish an exact $\mathcal{GP}$ baseline that made predictions without any approximation. We trained the $\mathcal{GP}$ hyperparameters using L-BFGS and early stopping on a validation dataset. For this to be feasible on large datasets, we used the Blackbox Matrix-Matrix multiplication inference procedure (Gardner et al., 2018; Wang et al., 2019). In the following sections, we discuss each method in detail.

**Variational $\mathcal{GP}$ approximation.** First, we compare the exact $\mathcal{GP}$ baseline with its variational approximation, i.e., $\mathcal{SVGP}$. First, consider the results on MPG and Concrete, where $\mathcal{SVGP}$ did not make use of inducing points due to the small sample size. Consequently, $\mathcal{SVGP}$ 's worse performance on Concrete is only due to the change of inference method. For the other datasets, we investigated the effect of the number of inducing points on the predictive log-likelihood; see Figure 11 in Appendix H. The dependence is very similar for all methods. In particular, the performance saturates at roughly 500 inducing points on all datasets except Kin40k, which continues to improve. However, the relative performance of the different methods on Kin40k is fairly stable.

**Elliptical likelihood.** Next, we consider whether it is advantageous to use an elliptical likelihood instead of a Gaussian. To this end, we compare the performance of $\mathcal{SVGP}$ and $\mathcal{EP}$-$\mathcal{GP}$, which only differ in this respect. The results show that switching to an elliptical likelihood improves the log-likelihood on most datasets, as would be expected theoretically.

**Elliptical posterior.** We now compare $\mathcal{EP}$-$\mathcal{GP}$ to $\mathcal{EP}$-$\mathcal{EP}$ to analyze the potential benefit of having an elliptical posterior. On three of the datasets (Bike, Kin40k, and Protein), which are all relatively large, the elliptical posterior produces a clear improvement in log-likelihood. In contrast, on the other datasets, it is similar, possibly because the posterior is well-approximated by a Gaussian. Regardless, we conclude that, when using an elliptical likelihood, an elliptical posterior is preferable over a Gaussian one.

**Heteroscedastic models.** Is there an additional benefit of having heteroscedastic noise? On the two smallest datasets (MPG and Concrete), the answer is clearly *no*: the heteroscedastic models perform worse than $\mathcal{SVGP}$ and $\mathcal{EP}$-$\mathcal{GP}$ in terms of both log-likelihood and mean squared error, indicating potential overfitting and that regularization may be warranted. (Note that the most relevant comparisons are Het-$\mathcal{GP}$ vs. $\mathcal{SVGP}$ and Het-$\mathcal{EP}$ vs. $\mathcal{EP}$-GP.)

On the remaining datasets, however, the heteroscedastic models clearly outperform $\mathcal{SVGP}$ and $\mathcal{EP}$-$\mathcal{GP}$ in terms of log-likelihood. On the other hand, they perform poorly in terms of mean squared error; in fact, worse than $\mathcal{SVGP}$ on all datasets. Hypothetically, this is because the heteroscedastic models attribute too much variation to the likelihood, thus sacrificing the mean-function prediction. This could potentially be mitigated by decoupling the mean and the covariance models (Salimbeni et al., 2018; Jankowiak et al., 2020). Another option would be to increase the weight of the KL-divergence term in the ELBO (Higgins et al., 2017). We expect such improvements to be more critical for the Het-$\mathcal{EP}$ model, which has a more flexible likelihood than Het-GP. Still, Het-$\mathcal{EP}$ already performs slightly better than Het-$\mathcal{GP}$ on the three largest datasets. Note, however, the $\mathcal{EP}$-$\mathcal{EP}$ model often achieves both a log-likelihood similar to the heteroscedastic models and a mean squared error similar to $\mathcal{SVGP}$.

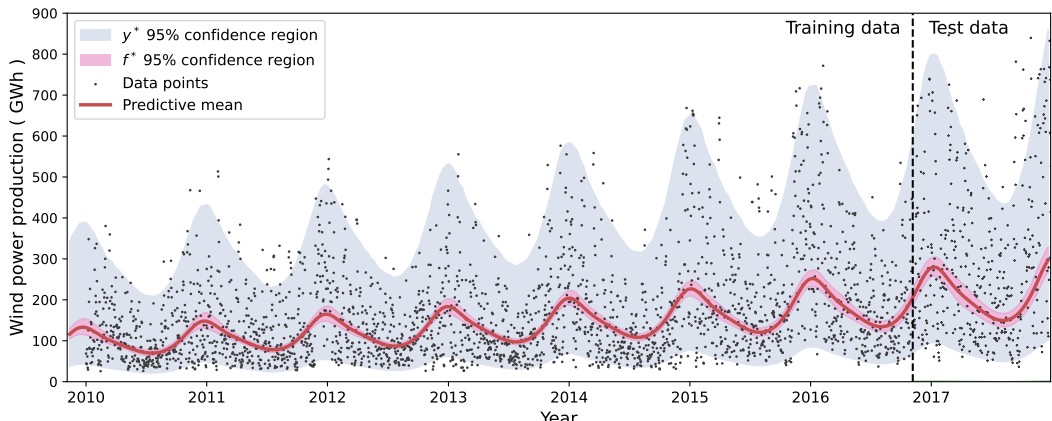

Figure 9: The predictive posterior after training the elliptical process on the wind power dataset. The shaded areas show the 95 % credibility area of the latent function posterior $f^*$ and the noisy posterior $y^*$.

**Computational considerations.** Empirically, we found that replacing the Gaussian likelihood with the elliptical likelihood had a minor impact on the computational demand. Further changing to an elliptical posterior increased the computational time per iteration, but a faster convergence partially compensated for this. Finally, modeling heteroscedastic noise with a neural network adds significant complexity, but this was offset by running it on a GPU.

**Prediction accuracy.** In summary, the results show that an elliptical likelihood results in better or equal predictive log-likelihoods than a Gaussian likelihood. However, the advantage is less significant on small datasets. Similarly, the more flexible elliptical posterior tends to produce better results. However, when looking at the mean squared error (MSE), the exact $\mathcal{GP}$ outperforms the other models. Thus, if predictive performance is the main objective, an exact $\mathcal{GP}$ (or, even better, a neural network) may be the best choice. However, the well-known scalability issues of exact $\mathcal{GP}$ clearly limit its applicability. In such scenarios, our results suggest that $\mathcal{EP}$-$\mathcal{EP}$ is a better choice than $\mathcal{SVGP}$.

### 4.4 Application: Forecasting wind power production

Wind power stands for a significant and increasing share of global power production. However, since wind power generation is inherently stochastic and hard to control it poses severe challenges to power system balance and stability (Impram et al., 2020). In principle, these could be addressed via sophisticated control theoretical tools such as stochastic model predictive control (Schildbach et al., 2014), but that requires accurate and reliable wind power forecasts. In this section, we illustrate the applicability of the elliptical process to time series forecasting. Specifically, we consider the task of making a probabilistic forecast of German country-wide totals of wind power production at the end of 2016 based on data from the preceding seven years. The data comes from Wiese et al. (2019).

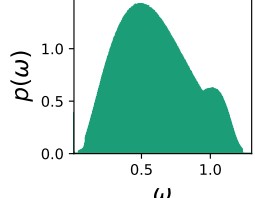

Figure 8: The mixing distribution $p(\omega)$ of the elliptical noise.

Since the data is strictly positive, we model the time series in the log domain, using an elliptical process with elliptical noise ($\mathcal{EP} - \mathcal{EP}$). To capture the periodicity in the data we use a sum of a periodic kernel and a linear kernel. See Appendix I for more details.

Figure 9 presents the predictive distribution from the elliptical process. The model successfully captures the underlying annual periodicity and slowly increasing trend while producing qualitatively reasonable credibility regions. Figure 8 shows the mixing distribution of the trained elliptical likelihood, which is rather broad and hence non-Gaussian. The two modes discernible in the mixing distribution reflect the seasonal changes in the underlying data. For comparison, Appendix I shows the corresponding results when using a full $\mathcal{GP}$ and

a heteroscedastic $\mathcal{EP}$, where the full $\mathcal{GP}$ seems to be more likely to overfit to short-scale trends in the data and produces credible regions that are too wide. The heteroscedastic $\mathcal{EP}$, on the other hand, produces a fit that is overall on par with the $\mathcal{EP}$ model but disentangles the seasonal variations.

### 4.5 Binary classification

To evaluate the $\mathcal{EP}$ on classification tasks, we perform variational $\mathcal{EP}$ and $\mathcal{GP}$ classification by simply replacing the likelihood with a binary one. To derive the expectation in Equation (18) we first sample $f_i \sim \mathcal{N}(f_i | \mu_{\boldsymbol{f}}(\boldsymbol{x}_i), \sigma_{\boldsymbol{f}}(\boldsymbol{x}_i)\xi)$ and then derive the likelihood as $\mathrm{Ber}(\mathrm{Sigmoid}(f_i))$.

This realization is interesting since we do not have a likelihood that captures the noise in the data; instead, the process itself has to do it. Therefore, we can indicate the value of the elliptical process itself without the elliptical noise. We compare two variational $\mathcal{EP}$ models with a variational $\mathcal{GP}$ model. The two $\mathcal{EP}$s differ in the prior mixing distributions, where the first model has a $\mathcal{GP}$ prior and a $\mathcal{EP}$ posterior. For the second model, we replace the $\mathcal{GP}$ prior to an elliptical one. We can see the trainable prior mixing distribution as using a continuously scaled mixture of Gaussian processes, which can be more expressive than a single $\mathcal{GP}$.

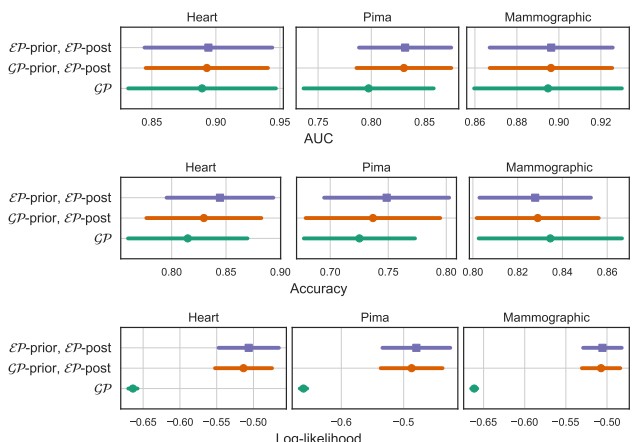

Figure 10: The classification AUC (Area Under the Curve), accuracy, and predictive log-likelihood from the ten-fold cross-validation (higher is better). We show the average of the ten splits as a dot and the standard deviation as a line.

To evaluate the models, we performed a ten-fold cross-validation where we trained the models on three classification datasets, described in Appendix J. Figure 10 presents the results from ten folds. From the area under the curve and the log-likelihood score, we see that the $\mathcal{EP}$ separates the two classes better, especially using the sparse models. The variational elliptical distribution is sufficient to get a high log-likelihood while training the mixing distribution of the $\mathcal{EP}$ did not further improve the score.

## 5 Related work

In general, attempts at modeling heavy-tailed stochastic processes modify either the likelihood or the stochastic process prior—rarely both. Approximate inference is typically needed when going beyond Gaussian likelihoods (Neal, 1997; Jylänki et al., 2011), e.g., for robust regression, but approximations that preserve analytical tractability have been proposed (Shah et al., 2014).

Elliptical processes gain flexibility by learning the mixing distribution, which makes training and inference reasonably efficient and the resulting predictions interpretable. It is, however, also possible to construct deep process priors by composing several stochastic process layers (Damianou & Lawrence, 2013) and including non-linear transformations as activation functions (Aitchison et al., 2021). The tractability of such deep processes depends on the specifics of the distributions involved. For instance, the deep inverse Wishart process (Aitchison et al., 2021) uses an inverse Wishart prior over the kernel just as the Student's $t$ process (Shah et al., 2014). This suggests it might be possible to generalize this approach to deep *elliptical* processes. While intriguing, we leave this to future work.

Other attempts at creating more expressive $\mathcal{GP}$ priors include Maroñas et al. (2021), who used a $\mathcal{GP}$ in combination with a normalizing flow, and Luo & Sun (2017), who used a discrete mixture of Gaussian processes. Similar ideas combining mixtures and normalizing flows have also been proposed to create more expressive likelihoods (Abdelhamed et al., 2019; Daemi et al., 2019; Winkler et al., 2019; Rivero & Dvorkin, 2020) and variational posteriors (Nguyen & Bonilla, 2014). Non-stationary extensions of Gaussian

processes, such as when modeling heteroscedastic noise, are somewhat rare. Examples include Zhao et al. (2021) who propose a hierarchical model in parameter space, the mixture model of Li et al. (2021), and the variational model of Lázaro-Gredilla & Titsias (2011).

Deep kernel learning (Calandra et al., 2016; Wilson et al., 2016b;a) is another class of deep $\mathcal{GP}$s that uses a neural network to learn the input features of a $\mathcal{GP}$. A similar approach was taken by Ma et al. (2019), who describe a class of stochastic processes where the finite-dimensional distributions are only defined implicitly as a parameterized transformation of some base distribution, thereby generalizing earlier work on warped Gaussian processes (Snelson et al., 2003; Rios & Tobar, 2019). However, the price of this generality is that standard variational inference is no longer possible. Based on an assumption of a Gaussian likelihood, they describe an alternative based on the wake-sleep algorithm by Hinton et al. (1995).

In the statistics literature, it is well-known that elliptical processes can be defined as scale-mixtures of Gaussian processes (Huang & Cambanis, 1979; O'Hagan, 1991; O'Hagan et al., 1999). However, unlike in machine learning, little emphasis is placed on building the models from data (i.e., training). These models have found applications in environmental statistics because of the field's inherent interest in modeling spatial extremes (Davison et al., 2012). Like us, several works take the mixing distribution as the starting point and make localized predictions of quantiles (Maume-Deschamps et al., 2017) or other tail-risk measures (Opitz, 2016).

## 6 Conclusions

The Gaussian distribution is the default choice in statistical modeling for good reasons. Even so, far from everything is Gaussian—casually pretending it is, comes at a risk. The elliptical distribution offers a computationally tractable alternative that can capture heavy-tailed distributions. The same reasoning applies when comparing the Gaussian process to the elliptical process. A sensible approach in many applications would be to start from the weaker assumptions of the elliptical process and let the data decide whether the evidence supports Gaussianity.

We constructed the elliptical process as a scale mixture of Gaussian distributions. By parameterizing the mixing distribution using a normalizing flow, we showed how a corresponding elliptical process can be trained using variational inference. The variational approximation we propose enables us to capture heavy-tailed posteriors and makes it straightforward to create a sparse variational elliptical process that scales to large datasets.

We performed both experiments on regression and classification. In particular, we investigated the benefits of various combinations of elliptical posterior and elliptical likelihood and their heteroscedastic counterparts. We concluded that using an elliptical likelihood and an elliptical posterior often achieves a better log-likelihood and similar mean squared error as the sparse variational $\mathcal{GP}$.

The added flexibility of elliptical processes could benefit a range of classical and new applications. However, advanced statistical models are not a cure-all, and one needs to avoid overreliance on such models, especially in safety-critical applications.

## 7 Acknowledgments

We thank Sebastian Mair and Zheng Zhao for providing feedback on the manuscript. We also want to thank the reviewers whose comments significantly improved the quality and clarity of our paper. This work was partially supported by the Wallenberg AI, Autonomous Systems and Software Program (WASP) funded by the Knut and Alice Wallenberg Foundation, the Swedish Foundation for Strategic Research grant SM19-0029, and by Kjell och Märta Beijer Foundation.

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

## A  The elliptical distribution

The Gaussian distribution—the basic building block of Gaussian processes—has several attractive properties that we wish the elliptical process to inherit, namely (i) closure under marginalization, (ii) closure under conditioning, and (iii) straightforward sampling. This leads us to consider the family of *consistent* elliptical distributions. Following Kano (1994), we say that a family of elliptical distributions $\{p(u(\boldsymbol{y}_N); \boldsymbol{\eta}) \,|\, N \in \mathbb{N}\}$ is consistent if and only if

$$\int_{-\infty}^{\infty} p\left(u(\boldsymbol{y}_{N+1}); \boldsymbol{\eta}\right) dy_{N+1} = p\left(u(\boldsymbol{y}_N); \boldsymbol{\eta}\right). \tag{21}$$

In other words, a consistent elliptical distribution is closed under marginalization.

Far from all elliptical distributions are consistent, but the complete characterization of those that are is provided by the following theorem (Kano, 1994).

**Theorem 1** *An elliptical distribution is consistent if and only if it originates from the integral*

$$p(u; \boldsymbol{\eta}) = |\boldsymbol{\Sigma}|^{-\frac{1}{2}} \int_0^{\infty} \left(\frac{1}{\xi 2\pi}\right)^{\frac{N}{2}} e^{\frac{-u}{2\xi}} p(\xi; \boldsymbol{\eta}_{\xi}) d\xi, \tag{22}$$

*where $\xi$ is a mixing variable with the corresponding, strictly positive finite, mixing distribution $p(\xi; \boldsymbol{\eta})$, that is independent of $N$.*

This shows that consistent elliptical distributions $p(u; \boldsymbol{\eta})$ are scale-mixtures of Gaussian distributions, with a mixing variable $\xi \sim p(\xi; \boldsymbol{\eta})$. Note that any mixing distribution fulfilling Theorem 1 can be used to define a consistent elliptical process. We recover the Gaussian distribution if the mixing distribution is a Dirac delta function and the Student's $t$ distribution if it is a scaled inverse chi-square distribution.

If $p(u; \boldsymbol{\eta})$ is a scale-mixture of normal distributions, it has the stochastic representation

$$\boldsymbol{Y} | \, \xi \sim \mathcal{N}(\boldsymbol{\mu}, \boldsymbol{\Sigma}\xi), \quad \xi \sim p(\xi; \boldsymbol{\eta}). \tag{23}$$

By using the following representation of the elliptical distribution,

$$\boldsymbol{Y} = \boldsymbol{\mu} + \boldsymbol{\Sigma}^{1/2} \boldsymbol{Z} \xi^{1/2}, \tag{24}$$

where $\boldsymbol{Z}$ follows the standard normal distribution, we get the mean

$$\mathbb{E}[\boldsymbol{Y}] = \boldsymbol{\mu} + \boldsymbol{\Sigma}^{1/2} \mathbb{E}[\boldsymbol{Z}] \ \mathbb{E}[\xi^{1/2}] = \boldsymbol{\mu} \tag{25}$$

and the covariance

$$\begin{aligned}
\text{Cov}(\boldsymbol{Y}) &= \mathbb{E}\left[(\boldsymbol{Y})\boldsymbol{\mu})(\boldsymbol{Y} - \boldsymbol{\mu})^{\top}\right] \\
&= \mathbb{E}\left[(\boldsymbol{\Sigma}^{1/2}\boldsymbol{Z}\sqrt{\xi})(\boldsymbol{\Sigma}^{1/2}\boldsymbol{Z}\sqrt{\xi})^{\top}\right] \\
&= \mathbb{E}\left[\xi\boldsymbol{\Sigma}^{1/2}\boldsymbol{Z}\boldsymbol{Z}^{\top}(\boldsymbol{\Sigma}^{1/2})^{\top}\right] \\
&= \mathbb{E}[\xi]\boldsymbol{\Sigma}.
\end{aligned} \tag{26}$$

The variance is a scale factor of the scale matrix $\boldsymbol{\Sigma}$. To get the variance we have to derive $\mathbb{E}[\xi]$. Note that if $\xi$ follows the scaled inverse chi-square distribution, $E[\xi] = \nu/(\nu - 2)$. We recognize this from the Student's $t$ distribution, where $\text{Cov}(\boldsymbol{Y}) = \nu/(\nu - 2)\boldsymbol{\Sigma}$.

## B  Conditional distribution

To use the $\mathcal{EP}$ for predictions, we need the conditional mean and covariance of the corresponding elliptical distribution, which we derive next. We partition the data as $\boldsymbol{y} = [\boldsymbol{y}_1, \boldsymbol{y}_2]$, where $\boldsymbol{y}_1$ are the $N_1$ observed data points, $\boldsymbol{y}_2$ are the next $N_2$ data points to predict, and $N_1 + N_2 = N$. We have the following result:

**Proposition 1** *If the data $\boldsymbol{y} = [\boldsymbol{y}_1, \boldsymbol{y}_2]$ originate from the consistent elliptical distribution in Equation (3), the conditional distribution originates from the distribution*

$$p_{\boldsymbol{y}_2|u_1}(\boldsymbol{y}_2) = \frac{c_{N_1,\boldsymbol{\eta}}}{\left|\boldsymbol{\Sigma}_{22|1}\right|^{\frac{1}{2}}(2\pi)^{\frac{N_2}{2}}} \int_0^\infty \xi^{-\frac{n}{2}} e^{-(u_{2|1}+u_1)\frac{1}{2\xi}} \, p(\xi; \boldsymbol{\eta})d\xi \tag{27}$$

*with the conditional mean $\mathbb{E}[\boldsymbol{y_2}|\boldsymbol{y}_1] = \boldsymbol{\mu}_{2|1}$ and the conditional covariance*

$$\mathrm{Cov}[\boldsymbol{Y}_2|\boldsymbol{Y}_1 = \boldsymbol{y}_2] = \mathbb{E}[\hat{\xi}]\boldsymbol{\Sigma}_{22|1}, \quad \hat{\xi} \sim \xi|\boldsymbol{y}_1, \tag{28}$$

*where $u_1 = (\boldsymbol{y}_1 - \boldsymbol{\mu}_1)^\top \boldsymbol{\Sigma}_{11}^{-1}(\boldsymbol{y}_1 - \boldsymbol{\mu}_1)$, $u_{2|1} = (\boldsymbol{y}_2 - \boldsymbol{\mu}_{2|1})^\top \Sigma_{22|1}^{-1}(\boldsymbol{y}_2 - \boldsymbol{\mu}_{2|1})$, and $c_{N_1,\boldsymbol{\eta}}$ is a normalization constant. The conditional scale matrix $\boldsymbol{\Sigma}_{22|1}$ and the conditional mean vector $\boldsymbol{\mu}_{2|1}$ are the same as the mean and the covariance matrix for a Gaussian distribution.*

The conditional distribution is guaranteed to be a consistent elliptical distribution but not necessarily the same as the original one—the shape depending on the training samples. (Recall that consistency only concerns the marginal distribution.)

**Proof of proposition 1.** The joint distribution of $[\boldsymbol{y}_1, \boldsymbol{y}_2]$ is $p(\boldsymbol{y}_1, \boldsymbol{y}_2|\xi)p(\xi; \boldsymbol{\eta})$ and the conditional distribution of $\boldsymbol{y}_2$, given $\boldsymbol{y}_1$ is $p(\boldsymbol{y}_2|\boldsymbol{y}_1, \xi)p(\xi|\boldsymbol{y}_1; \boldsymbol{\eta})$.

For a given $\xi$, $p(\boldsymbol{y}_2|\boldsymbol{y}_1, \xi)$ is the conditional normal distribution and so

$$p(\boldsymbol{y}_2|\boldsymbol{y}_1, \xi) \sim \mathcal{N}(\boldsymbol{\mu}_{2|1}, \Sigma_{22|1}\hat{\xi}), \quad \hat{\xi} \sim p(\xi|\boldsymbol{y}_1; \boldsymbol{\eta}) \tag{29}$$

where,

$$\boldsymbol{\mu}_{2|1} = \boldsymbol{\mu}_2 + \Sigma_{21}\Sigma_{11}^{-1}(\boldsymbol{y}_1 - \boldsymbol{\mu}_1) \tag{30}$$

$$\Sigma_{22|1} = \Sigma_{22} - \Sigma_{21}\Sigma_{11}^{-1}\Sigma_{21}, \tag{31}$$

the same as for the conditional Gaussian distribution. We obtain the conditional distribution $p(\xi|\boldsymbol{y}_1; \boldsymbol{\eta})$ by remembering that

$$p(\boldsymbol{y}_1|\xi) \sim \mathcal{N}(\boldsymbol{\mu}_1, \Sigma_{11}\xi). \tag{32}$$

Using Bayes' Theorem we get

$$p(\xi|\boldsymbol{y}_1; \boldsymbol{\eta}) \propto p(\boldsymbol{y}_1|\xi)p(\xi; \boldsymbol{\eta})$$

$$\propto |\Sigma_{11}\xi|^{-1/2} \exp\left\{-\frac{u_1}{2\xi}\right\} p(\xi; \boldsymbol{\eta})$$

$$\propto \xi^{-N_1/2} \exp\left\{-\xi\frac{u_1}{2}\right\} p(\xi; \boldsymbol{\eta}). \tag{33}$$

Recall that $u_1 = (\boldsymbol{y} - \boldsymbol{\mu}_1)^\top \boldsymbol{\Sigma}_{11}^{-1}(\boldsymbol{y} - \boldsymbol{\mu}_1))$. We normalize the distribution by

$$c_{N_1,\boldsymbol{\eta}}^{-1} = \int_0^\infty \xi^{-N_1/2} \exp\left\{-\frac{u_1}{2\xi}\right\} p(\xi; \boldsymbol{\eta})d\xi. \tag{34}$$

The conditional mixing distribution is

$$p(\xi|\boldsymbol{y}_1; \boldsymbol{\eta}) = c_{N_1,\boldsymbol{\eta}}\xi^{-N_1/2} \exp\left\{-\frac{u_1}{2\xi}\right\} p(\xi; \boldsymbol{\eta}). \tag{35}$$

The conditional distribution of $\boldsymbol{y}_2$ given $\boldsymbol{y}_1$ is derived by using the consistency formula

$$p(\boldsymbol{y}_2|\boldsymbol{y}_1) = \frac{1}{|\boldsymbol{\Sigma}_{22|1}|^{1/2}(2\pi)^{N_2/2}} \int_0^\infty \xi^{-N_2/2} \exp{-\frac{u_{2|1}}{2\xi}} p(\xi|\boldsymbol{y}_1)d\xi, \tag{36}$$

where $u_{2|1} = (\boldsymbol{y}_2 - \boldsymbol{\mu}_{2|1})^\top \Sigma_{22|1}^{-1}(\boldsymbol{y}_2 - \boldsymbol{\mu}_{2|1})$. Using Equation (35) we get

$$p(\boldsymbol{y}_2|\boldsymbol{y}_1) = \frac{c_{N_1,\boldsymbol{\eta}}}{|\boldsymbol{\Sigma}_{22|1}|^{1/2}(2\pi)^{N_2/2}} \int_0^\infty \xi^{-n/2} e^{-(u_{2|1}+u_1)/(2\xi)} p(\xi; \boldsymbol{\eta})d\xi. \tag{37}$$

## C  Derivation of the credible intervals of the elliptical process

We derive the credible interval of the elliptical process, by using the Monte Carlo approximation of the integral, as

$$p(-z\sigma < x < z\sigma) = \frac{1}{\sigma\sqrt{2\pi}} \int_{-z\sigma}^{z\sigma} \int_0^\infty \xi^{-1/2} e^{-x^2/(\xi 2\sigma^2)} p(\xi) d\xi dx \tag{38}$$

$$= \frac{1}{\sigma\sqrt{2\pi}} \int_{-z\sigma}^{z\sigma} \frac{1}{m} \sum_{i=1}^m \xi_i^{-1/2} e^{-x^2/(2\xi_i\sigma^2)} dx \tag{39}$$

$$= \frac{1}{\sigma m\sqrt{2\pi}} \sum_{i=1}^m \xi_i^{-1/2} \int_{-z\sigma}^{z\sigma} e^{-x^2/(2\xi_i\sigma^2)} dx \tag{40}$$

$$= \frac{2}{m\sqrt{\pi}} \sum_{i=1}^m \int_0^{\frac{z}{\sqrt{2\xi_i}}} e^{-u^2} du \tag{41}$$

$$= \frac{1}{m} \sum_{i=1}^m \operatorname{erf}\left(\frac{z}{\sqrt{2\xi_i}}\right). \tag{42}$$

For every mixing distribution, we can derive the credibility of the prediction. It is the number of samples $m$ we take that decides the accuracy of the credible interval.

## D  Details on the non-sparse variational elliptical process

For a Gaussian process, the posterior of the latent variables $\boldsymbol{f}$ is

$$p(\boldsymbol{f}|\boldsymbol{y}) \propto p(\boldsymbol{y}|\boldsymbol{f})p(\boldsymbol{f}). \tag{43}$$

Here, the prior $p(\boldsymbol{f}|\boldsymbol{X}) \sim \mathcal{N}(0, \boldsymbol{K})$ is a Gaussian process with kernel $\boldsymbol{K}$ and the likelihood $p(\boldsymbol{y}|\boldsymbol{f}) \sim \mathcal{N}(\boldsymbol{f}, \sigma^2\boldsymbol{I})$ is Gaussian. The posterior derives to

$$p(\boldsymbol{f}|\boldsymbol{y}) \sim \mathcal{N}\left(\boldsymbol{f}|\boldsymbol{K}\left(\boldsymbol{K} + \sigma^2 I\right)^{-1}\boldsymbol{y}, \left(\boldsymbol{K}^{-1} + \sigma^{-2}\boldsymbol{I}\right)^{-1}\right) \tag{44}$$

and the predictive distribution of an arbitrary input location $x^*$ is

$$p(f^*|\boldsymbol{y}) = \int p(f^*|\boldsymbol{f})p(\boldsymbol{f}|\boldsymbol{y})d\boldsymbol{f}, \tag{45}$$

where $p(f^*|\boldsymbol{f}, \boldsymbol{x}, \boldsymbol{x}^*)$ is the conditional distribution, which is again Gaussian with

$$\mathcal{N}\left(f^*|\boldsymbol{k}_*^\top\left(\boldsymbol{k} + \sigma^2\boldsymbol{I}\right)^{-1}\boldsymbol{y}, k_{**} - \boldsymbol{k}_*^\top\left(\boldsymbol{K} + \sigma^2\boldsymbol{I}\right)^{-1}\boldsymbol{k}_*\right). \tag{46}$$

Going back to the elliptical process, we want to derive the predictive distribution. The problem, though, is that the posterior is now intractable. In order to get a tractable posterior, we train the model using variational inference, where we approximate the intractable posterior with a tractable one,

$$p(\boldsymbol{f}, \xi|\boldsymbol{y}; \boldsymbol{\eta_f}, \boldsymbol{\eta_\xi}) \approx q(\boldsymbol{f}, \xi; \boldsymbol{\varphi_f}, \boldsymbol{\varphi_\xi}) = q(\boldsymbol{f}|\xi; \boldsymbol{\varphi_f})q(\xi; \boldsymbol{\varphi_\xi}). \tag{47}$$

Here $\boldsymbol{\eta_f}$ are the parameters of the prior $\mathcal{GP}$ process (such as the kernel parameters), $\boldsymbol{\eta_\xi}$ are the parameters of the mixing distribution, and $q(\boldsymbol{f}|\xi; \boldsymbol{\varphi_f}) \sim \mathcal{N}(\boldsymbol{m_f}, \boldsymbol{S_f}\xi)$, where $\boldsymbol{m_f}$ and $\boldsymbol{S_f}$ are variational parameters. The posterior $q(\xi; \boldsymbol{\varphi_\xi})$ is parameterized with any positive distribution such as a normalizing flow. We use this approximation when we derive the predictive distribution

$$p(f^*|\boldsymbol{y}) = \int p(f^*|\boldsymbol{f}, \xi; \boldsymbol{\eta_f})p(\boldsymbol{f}, \xi|\boldsymbol{y}; \boldsymbol{\eta_f}, \boldsymbol{\eta_\xi})d\boldsymbol{f}d\xi \tag{48}$$

$$= \int p(f^*|\boldsymbol{f}, \xi; \boldsymbol{\eta_f})p(\boldsymbol{f}, \xi|\boldsymbol{y}; \boldsymbol{\eta_f}, \boldsymbol{\eta_\xi})d\boldsymbol{f}d\xi \tag{49}$$

$$\approx \int p(f^*|\boldsymbol{f}, \xi; \boldsymbol{\eta_f})q(\boldsymbol{f}|\xi; \boldsymbol{\varphi_f})q(\xi; \boldsymbol{\varphi_\xi})d\boldsymbol{f}d\xi. \tag{50}$$

By first taking a look at the prior distribution $p(f^*, \boldsymbol{f}|\xi)$ when $\xi$ is constant,

$$\begin{bmatrix} f^* \\ \boldsymbol{f} \end{bmatrix} \xi \sim \mathcal{N}\left( 0, \begin{bmatrix} k_{**} & \boldsymbol{k}_*^\top \\ \boldsymbol{k}_* & \boldsymbol{K} \end{bmatrix} \xi \right), \tag{51}$$

we arrive at the conditional distribution

$$p(f^*|\boldsymbol{f}, \xi; \boldsymbol{\eta_f}) = \mathcal{N}\left( \boldsymbol{k}_*^\top \boldsymbol{K}^{-1} \boldsymbol{f}, \left( k_{**} - \boldsymbol{k}_*^\top \boldsymbol{K}^{-1} \boldsymbol{k}_* \right) \xi \right). \tag{52}$$

We use this expression together with the variational approximation to derive the posterior predictive distribution

$$p(f^*|\boldsymbol{y}) = \int p(f^*|\boldsymbol{f}, \xi; \boldsymbol{\eta_f}) q(\boldsymbol{f}|\xi; \boldsymbol{\varphi_f}) q(\xi; \boldsymbol{\varphi_\xi}) d\boldsymbol{f} d\xi \tag{53}$$

$$= \mathbb{E}_{q(\xi; \boldsymbol{\varphi_\xi})} \left[ \int p(f^*|\boldsymbol{f}, \xi; \boldsymbol{\eta_f}) q(\boldsymbol{f}|\xi; \varphi_f) d\boldsymbol{f} \right] \tag{54}$$

$$= \mathbb{E}_{q(\xi; \boldsymbol{\varphi_\xi})} \left[ \int \mathcal{N}\left( f^*|\boldsymbol{k}_*^\top \boldsymbol{K}^{-1} \boldsymbol{f}, (k_{**} - \boldsymbol{k}_*^\top \boldsymbol{K}^{-1} \boldsymbol{k}_*)\xi \right) \mathcal{N}\left( \boldsymbol{f}|\boldsymbol{m}, \boldsymbol{S}\xi \right) d\boldsymbol{f} \right] \tag{55}$$

$$= \mathbb{E}_{q(\xi; \boldsymbol{\varphi_\xi})} \left[ \mathcal{N}(f^*|\mu_{\boldsymbol{f}}(\boldsymbol{x}^*), \sigma_{\boldsymbol{f}}(\boldsymbol{x}^*)\xi) \right], \tag{56}$$

where

$$\mu_{\boldsymbol{f}}(\boldsymbol{x}^*) = \boldsymbol{k}_*^\top \boldsymbol{K}^{-1} \boldsymbol{m}, \tag{57}$$

$$\sigma_{\boldsymbol{f}}(\boldsymbol{x}^*) = k_{**} - \boldsymbol{k}_*^\top \left( \boldsymbol{K}^{-1} - \boldsymbol{K}^{-1} \boldsymbol{S} \boldsymbol{K}^{-1} \right) \boldsymbol{k}_*. \tag{58}$$

We get the variance by $\mathbb{E}[\xi]\sigma_{\boldsymbol{f}}(\boldsymbol{x}^*)$.

**Optimizing the ELBO**  We train the model by optimizing the evidence lower bound (ELBO) given by

$$\mathcal{L}_{\mathrm{ELBO}}(\boldsymbol{\eta_f}, \boldsymbol{\eta_\xi}, \boldsymbol{\varphi_f}, \boldsymbol{\varphi_\xi}) = \mathbb{E}_{q(\boldsymbol{f}, \xi; \boldsymbol{\varphi})} \left[ \log p(\boldsymbol{y}|\boldsymbol{f}) \right] - D_{\mathrm{KL}}\left( q(\boldsymbol{f}, \xi; \boldsymbol{\varphi_f}, \boldsymbol{\varphi_\xi}) \,||\, p(\boldsymbol{f}, \xi; \boldsymbol{\eta_f}, \boldsymbol{\eta_\xi}) \right). \tag{59}$$

# E  Details on sparse elliptical processes

With the variational inference framework, we create a sparse version of the model

$$\int p(\boldsymbol{f}, \boldsymbol{u}, \xi; \boldsymbol{\eta}) d\xi = \int p(\boldsymbol{f}|\boldsymbol{u}, \xi; \boldsymbol{\eta_f}) p(\boldsymbol{u}|\xi; \boldsymbol{\eta_u}) p(\xi; \boldsymbol{\eta_\xi}) d\xi, \tag{60}$$

where $\boldsymbol{u}$ are outputs of the elliptical process located at the inducing inputs $\boldsymbol{Z}$. We approximate the posterior with

$$p(\boldsymbol{f}, \boldsymbol{u}, \xi|\boldsymbol{y}; \boldsymbol{\eta}) \approx p(\boldsymbol{f}|\boldsymbol{u}, \xi; \boldsymbol{\eta_f}) q(\boldsymbol{u}|\xi; \boldsymbol{\varphi_u}) q(\xi; \boldsymbol{\varphi_\xi}). \tag{61}$$

The posterior predictive distribution is then given by

$$p(f^*|\boldsymbol{y}) = \int p(f^*|\boldsymbol{f}, \boldsymbol{u}, \xi; \boldsymbol{\eta}) p(\boldsymbol{f}, \boldsymbol{u}, \xi|\boldsymbol{y}; \boldsymbol{\eta}) d\boldsymbol{f} d\boldsymbol{u} d\xi$$

$$\approx \int p(f^*|\boldsymbol{f}, \boldsymbol{u}, \xi; \boldsymbol{\eta}) p(\boldsymbol{f}|\boldsymbol{u}, \xi; \boldsymbol{\eta_f}) q(\boldsymbol{u}|\xi; \boldsymbol{\varphi_u}) q(\xi; \boldsymbol{\varphi_\xi}) d\boldsymbol{f} d\boldsymbol{u} d\xi$$

$$= \int \left[ \int p(f^*|\boldsymbol{f}, \boldsymbol{u}, \xi; \boldsymbol{\eta}) p(\boldsymbol{f}|\boldsymbol{u}, \xi; \boldsymbol{\eta_f}) d\boldsymbol{f} \right] q(\boldsymbol{u}|\xi; \boldsymbol{\varphi_u}) q(\xi; \boldsymbol{\varphi_\xi}) d\boldsymbol{u} d\xi. \tag{62}$$

We can simplify the inner expression by using the fact that the elliptical distribution is consistent,

$$\int p(f^*|\boldsymbol{f}, \boldsymbol{u}, \xi; \boldsymbol{\eta}) p(\boldsymbol{f}|\boldsymbol{u}, \xi; \boldsymbol{\eta}) d\boldsymbol{f} = \int p(f^*, \boldsymbol{f}|\boldsymbol{u}, \xi; \boldsymbol{\eta}) d\boldsymbol{f} = p(f^*|\boldsymbol{u}, \xi; \boldsymbol{\eta}). \tag{63}$$

Hence, Equation (62) is simplifies to

$$p(f^*|\boldsymbol{y}) = \int p(f^*|\boldsymbol{u}, \xi; \boldsymbol{\eta})q(\boldsymbol{u}|\xi; \boldsymbol{\varphi_u})q(\xi; \boldsymbol{\varphi_\xi})d\boldsymbol{u}d\xi, \tag{64}$$

where $q(\boldsymbol{u}|\xi; \boldsymbol{\varphi_u}) = \mathcal{N}(\boldsymbol{m}_u, \boldsymbol{S}_u\xi)$ with the variational parameters $\boldsymbol{m}_u$ and $\boldsymbol{S}_u$, and $\xi$ is parameterized, e.g., by a normalizing flow.

Finally, we obtain the posterior $p(f^*|\boldsymbol{x}^*) = \mathbb{E}_{q(\xi;\boldsymbol{\varphi_\xi})}\left[\mathcal{N}(f^*|\mu_{\boldsymbol{f}}(\boldsymbol{x}^*), \xi\sigma_{\boldsymbol{f}}(\boldsymbol{x}^*))\right]$, where

$$\mu_{\boldsymbol{f}}(\boldsymbol{x}^*) = \boldsymbol{k}_*^\top \boldsymbol{K_{uu}}^{-1}\boldsymbol{m} \tag{65}$$

$$\sigma_{\boldsymbol{f}}(\boldsymbol{x}^*) = k_{**} - \boldsymbol{k}_i^\top \left(\boldsymbol{K_{uu}}^{-1} - \boldsymbol{K_{uu}}^{-1}\boldsymbol{S}\boldsymbol{K_{uu}}^{-1}\right)\boldsymbol{k}_*. \tag{66}$$

Here, $\boldsymbol{k}_* = k(\boldsymbol{x}^*, \boldsymbol{Z})$, $k_{**} = k(\boldsymbol{x}^*, \boldsymbol{x}^*)$, and $\boldsymbol{K_{uu}} = k(\boldsymbol{Z}, \boldsymbol{Z})$.

## F  Implementation: variational inference

We used the Pyro library (Bingham et al., 2019), a universal probabilistic programming language (PPL) written in Python and supported by PyTorch on the backend.

In Pyro, we trained a model with variational inference (Kingma & Welling, 2014) by creating "stochastic functions" called **model** and a **guide**, where the **model** samples from the prior latent distributions $p(\boldsymbol{f}, \xi, \omega; \boldsymbol{\eta})$, and the observed distribution $p(\boldsymbol{y}|\boldsymbol{f}, \omega)$, and the **guide** samples the approximate posterior $q(\boldsymbol{f}|\xi; \boldsymbol{\varphi_f})q(\xi; \boldsymbol{\varphi_\xi})$. We then trained the model by maximizing the evidence lower bound (ELBO), where we simultaneously optimized the model parameters $\boldsymbol{\eta}$ and the variational parameters $\boldsymbol{\varphi}$. (See more details here.)

To implement the model in Pyro, we created the guide and the model (see Algorithm 1) by building upon the already implemented variational Gaussian process. We used the guide and the model to derive the ELBO, which we then optimized with stochastic gradient descent using the Adam optimizer (Kingma & Ba, 2015).

---

**Algorithm 1** PyTorch implementation of the variational sparse elliptical process (VI-$\mathcal{EP}$-$\mathcal{EP}$).

1: **procedure** MODEL($\boldsymbol{X}, \boldsymbol{y}$)
2:     Sample $\xi =$ from $p(\xi; \boldsymbol{\eta}_\xi)$( Normalizing flow )
3:     Sample $\boldsymbol{u}$ from $\mathcal{N}(\boldsymbol{0}, \xi\boldsymbol{K_{uu}}))$                    ▷ Take a sample from the latent $\boldsymbol{u}$ and $\xi$
4:     Derive the variational posterior $\prod_{i=1}^{N} q(f_i|\xi; \boldsymbol{\varphi}) = \mathcal{N}(\mu_{\boldsymbol{f}}(\boldsymbol{x}_i), \sigma_{\boldsymbol{f}}(\boldsymbol{x}_i)\xi)$. ▷ During training $\xi$ is sampled from the posterior/guide.
5:     Take a Monte-Carlo sample $\hat{f}_i$ from each $q(f_i|\xi; \varphi)$
6:     For or each $y_i$ approximate $\ell_{y_i} = \log \int \mathcal{N}(y_i|f_i, \omega) p(\omega; \boldsymbol{\eta}_\omega)d\omega$ using the trapezoid rule.
7:     Get the log probability of $\boldsymbol{y}$ by $\sum_{i=1}^{N} \ell_{y_i}$.
8: **end procedure**
9: **procedure** GUIDE
10:     Sample $\xi =$ from $q(\xi; \boldsymbol{\varphi}_\xi)$( Normalizing flow )
11:     Sample $\boldsymbol{u}$, from $\mathcal{N}(\boldsymbol{m}, \boldsymbol{S}\xi)$
12: **end procedure**

---

## G  Regression experiment setup

In the regression experiments in Section 4.3, we ran all experiments using the Adam optimizer (Kingma & Ba, 2015) with a learning rate of 0.01. For the full $\mathcal{GP}$, we used the L-BFGS optimizer to train the hyperparameters. Here, we, in the same way as for the other models, used early stopping on a validation dataset, which operated by saving the model with the lowest validation log predictive likelihood.

For all experiments, we created ten random train/val/test splits with the proportions 0.6/0.2/0.2, except for the two smallest datasets (mpg and concrete), where we neglected the validation dataset and used a

train/test proportions of 0.7/0.3. For the test set evaluation, we used the model with the highest predictive probability on the validation set. For the large datasets ($N > 1000$), we used 500 inducing points. We did not use a sparse version of the model for the small datasets but instead used $\boldsymbol{Z} = \boldsymbol{X}_{train}$ and kept them fixed during the training. We run the optimizer for the large dataset for 500 epochs and the small dataset for 2000 epochs.

**Elliptical process setup.** The likelihood mixing distribution uses a spline flow with nine bins and *Softplus* as its output transformation. The elliptic posterior mixing distribution uses a spline flow with five bins and a *Sigmoid* output transformation. These parameters were not tuned, and fixed for all experiments.

For the heteroscedastic noise models, we used two-layer neural networks with hidden dimensions of 128. For the elliptical noise, we learned a spline flow with nine bins which results in 26 hyperparameters to learn while for the heteroscedastic Gaussian likelihood we learned the variance solely.

## H    Results

The regression results from Figure 7 are presented in Tables 2 and 3. Figure 11 presents the outcome for different numbers of inducing points. We see that the results have stabilized for almost all datasets at 500 inducing points. We also notice that the relative log-likelihood between the models stays constant after 400-500 inducing points.

Table 2: Predictive mean squared error (MSE) on the hold-out sets from the experiments. We show the average of the ten random splits and one standard deviation in parenthesis.

|  | MPG | CONCRETE | ELEVATORS | BIKE | CALIFORNIA | KIN40K | PROTEIN |
|---|---|---|---|---|---|---|---|
| Het-$\mathcal{EP}$ | 0.144 (0.018) | 0.127 (0.014) | 0.149 (0.005) | 0.018 (0.002) | 0.223 (0.012) | 0.141 (0.008) | 0.531 (0.009) |
| Het-$\mathcal{GP}$ | 0.142 (0.020) | 0.178 (0.022) | 0.148 (0.005) | 0.020 (0.002) | 0.230 (0.011) | 0.112 (0.007) | 0.508 (0.007) |
| EP-$\mathcal{EP}$ | 0.122 (0.027) | 0.176 (0.022) | 0.145 (0.005) | 0.007 (0.001) | 0.223 (0.012) | 0.042 (0.002) | 0.433 (0.008) |
| EP-$\mathcal{GP}$ | 0.121 (0.026) | 0.128 (0.013) | 0.145 (0.005) | 0.011 (0.001) | 0.226 (0.013) | 0.056 (0.003) | 0.481 (0.007) |
| $\mathcal{SVGP}$ | 0.122 (0.018) | 0.128 (0.011) | 0.143 (0.004) | 0.007 (0.001) | 0.219 (0.012) | 0.047 (0.001) | 0.477 (0.007) |
| Exact $\mathcal{GP}$ | 0.135 (0.135) | 0.103 (0.103) | 0.134 (0.134) | 0.002 (0.002) | 0.134 (0.134) | 0.006 (0.000) | 0.357 (0.006) |

Table 3: Negative log likelihood (Neg LL) on the hold-out test sets from the experiments. We show the average of the ten random splits and one standard deviation in parenthesis.

|  | MPG | CONCRETE | ELEVATORS | BIKE | CALIFORNIA | KIN40K | PROTEIN |
|---|---|---|---|---|---|---|---|
| Het-$\mathcal{EP}$ | 0.463 (0.089) | 0.332 (0.033) | 0.400 (0.018) | -1.327 (0.020) | 0.506 (0.022) | -0.397 (0.012) | 0.921 (0.017) |
| Het-$\mathcal{GP}$ | 0.530 (0.155) | 0.456 (0.074) | 0.399 (0.017) | -1.532 (0.047) | 0.376 (0.021) | -0.316 (0.010) | 0.986 (0.013) |
| EP-$\mathcal{EP}$ | 0.266 (0.074) | 0.443 (0.083) | 0.425 (0.015) | -1.406 (0.028) | 0.506 (0.022) | -0.246 (0.020) | 0.976 (0.008) |
| EP-$\mathcal{GP}$ | 0.268 (0.071) | 0.344 (0.031) | 0.427 (0.014) | -1.304 (0.023) | 0.515 (0.021) | -0.053 (0.049) | 1.056 (0.006) |
| $\mathcal{SVGP}$ | 0.352 (0.062) | 0.382 (0.030) | 0.446 (0.013) | -0.865 (0.016) | 0.649 (0.022) | -0.028 (0.004) | 1.056 (0.006) |
| Exact $\mathcal{GP}$ | 0.387 (0.387) | 0.206 (0.206) | 0.463 (0.463) | -1.103 (-1.103) | 0.463 (0.463) | -0.166 (0.097) | 0.970 (0.005) |

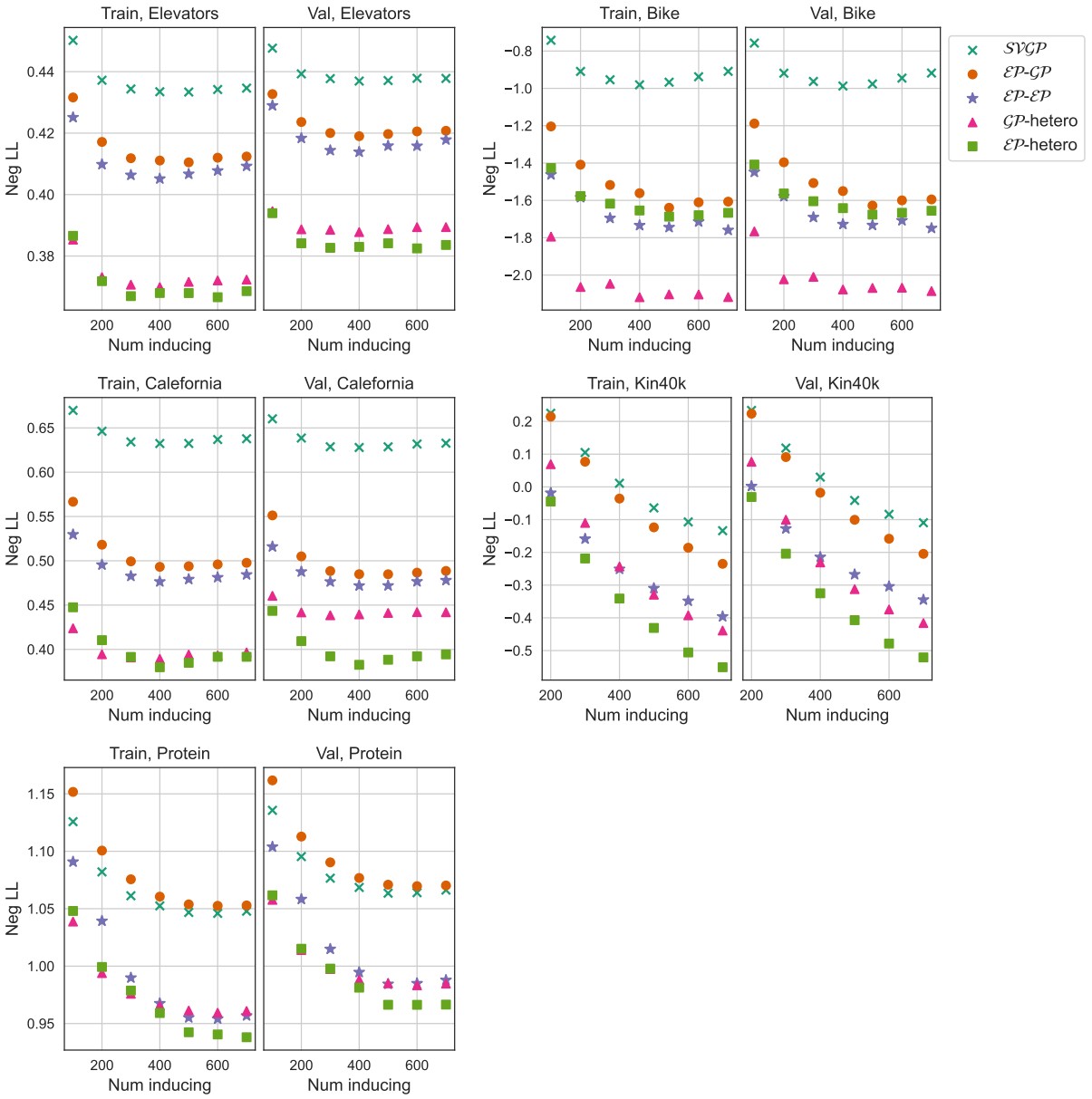

Figure 11: The train and validation negative log-likelihood for the datasets using a varying number of inducing points.

# I  Application: Wind power production

In the wind power production experiment, we used an $\mathcal{EP}$ process with elliptical likelihood, elliptical posterior, and 300 inducing points; a heteroscedastic $\mathcal{EP}$ with heteroscedastic elliptical likelihood, and elliptical posterior; and a full $\mathcal{GP}$ with Gaussian likelihood. They all used a kernel that was a sum of a periodic kernel and a linear kernel. Before training, we transformed to data to log scale and then normalized it. See Figure 12 for a visualization of the data before and after the transformation. This transformation made the data more symmetric around its mean and removed the nonnegative constraints.

Figure 9 in Section 4.4 shows the results when training the data with an $\mathcal{EP}$ process with elliptical likelihood. For comparison, we here plot the results when using a full $\mathcal{GP}$ (Figure 13), and heteroscedastic $\mathcal{EP}$ (Figure 14).

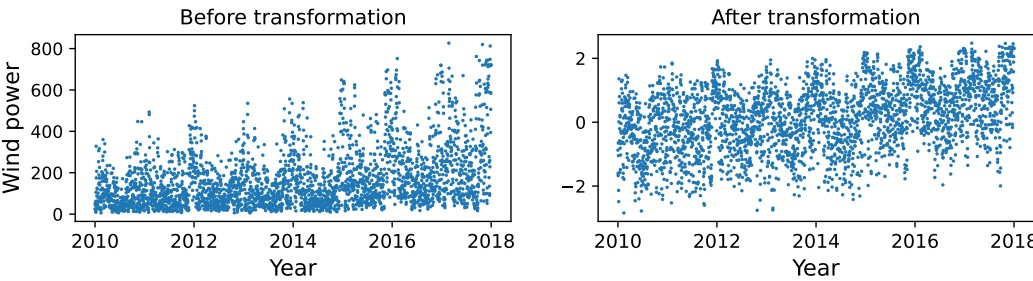

Figure 12: The data used in the wind power production experiments, where the left plot illustrates the data before it is transformed and the right plot shows the data after it is transformed, which is what we used as input to the model.

The main takeaway is that all three models are able to capture the periodicity in the data. The full $\mathcal{GP}$ seems to have a tendency to overfit the data and also to create wide credibility regions. This is probably because the data is non-Gaussian and the thin tails of the $\mathcal{GP}$ forces variances to be high. The heteroscedastic $\mathcal{EP}$ fits the data well, although the difference from using a non-heteroscedastic likelihood is small. Figure 15 illustrates the mixing distribution during some of the months of the year where we clearly can see that we have a seasonal change in the noise.

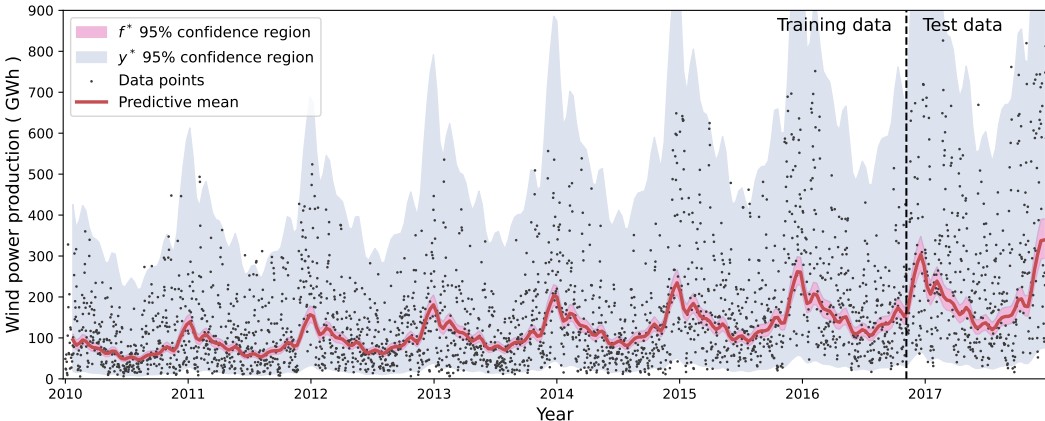

Figure 13: Wind power forecast using an exact $\mathcal{GP}$ The plot shows the predictive posterior where the shaded areas show the 95 % credibility area of the latent function posterior $f^*$ and the noisy posterior $y^*$.

## J  Datasets

**Bike dataset**   (Fanaee-T & Gama, 2014) is obtained from bike sharing data, especially it contains the hourly and daily count of rental bikes between the years 2011 and 2012 with the corresponding weather and seasonal information.

**Elevators dataset**   (Dua & Graff, 2017) is obtained from the task of controlling an F16 aircraft, and the objective is related to an action taken on the elevators of the aircraft according to the status attributes of the airplane.

**Physicochemical properties of protein tertiary structure dataset**   The dataset is taken from CASP 5-9. There are 45730 decoys and sizes varying from 0 to 21 Armstrong.

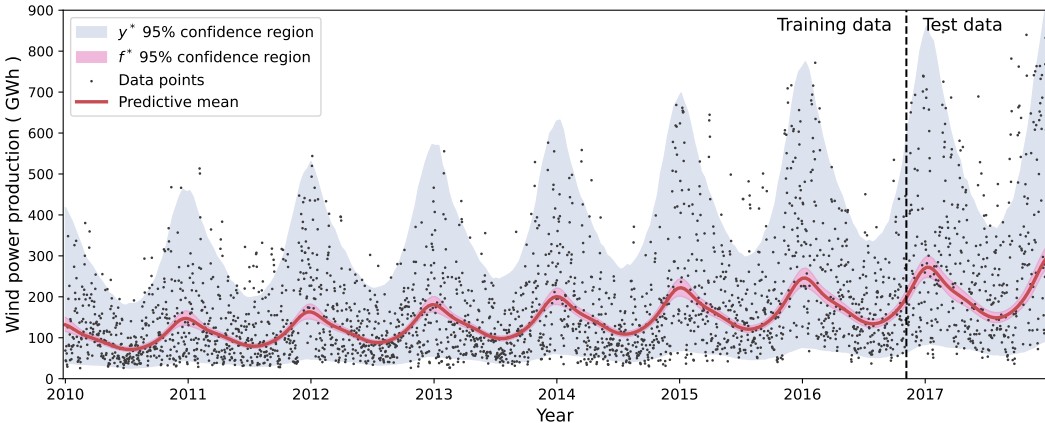

Figure 14: The predictive posterior after training the elliptical process with heteroscedastic noise on the wind power dataset. The shaded areas show the 95 % credibility area of the latent function posterior $f^*$ and the noisy posterior $y^*$.

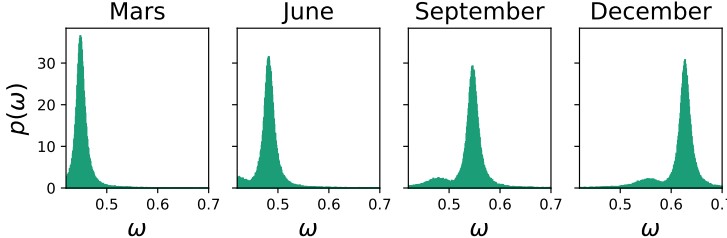

Figure 15: The mixing distribution of the elliptical heteroscedastic likelihood sampled during different months during the year.

**California housing dataset** was originally published by Pace & Barry (1997). There are 20 640 samples and 9 feature variables in this dataset. The targets are the prices of houses in the California area.

**The Concrete dataset** (Yeh, 1998) has 8 input variables and 1030 observations. The target variables are the concrete compressive strength.

**Auto MPG dataset** (Alcalá-Fdez et al., 2011) originally from the StatLib library which is maintained at Carnegie Mellon University. The data concerns city-cycle fuel consumption in miles per gallon and consists of 392 samples with five features each.

**Pima Indians Diabetes Database** (Smith et al., 1988) originally from the National Institute of Diabetes and Digestive and Kidney Diseases. The objective of the dataset is to predict diagnostically whether or not a patient has diabetes based on certain diagnostic measurements included in the dataset. The dataset consists of 768 samples with eight attributes.

**The Cleveland Heart Disease dataset** consists of 13 input variables and 270 samples. The target classifies whether a person is suffering from heart disease or not.

**The Mammography Mass dataset** predicts the severity (benign or malignant) of a mammography mass lesion from BI-RADS attributes and the patient's age. This dataset consists of 961 with six attributes.

