# OpenReview forum: "Variational Elliptical Processes"
_TMLR — Accepted by TMLR_

### Review · Reviewer_g9nQ · 2023-04-07

**Summary Of Contributions:**

In this work, the authors present a novel interpretation of elliptical processes that leverages variational inference in order to approximate intractable likelihoods and reduce computational complexity. This class of models is expected to outperform Gaussian processes (GPs) on datasets having heavier-tailed distributions, while also subsuming the former along with Student-$t$ processes. Via an extensive combination of experiments on synthetic and real-world datasets, the authors demonstrate how different aspects of variational $\mathcal{EP}$s (i.e. their likelihood, posterior approximation, etc) contribute towards the overall performance of the model in comparison to the more widely-used GPs.

**Audience:**

Yes

**Broader Impact Concerns:**

I do not foresee any broader impact concerns resulting from this work.

**Claims And Evidence:**

Yes

**Requested Changes:**

The main changes I would like to see in a future revision of the paper are closely tied to the weakness listed in the previous section. For clarity, these are:

- *[major]* Clarifying whether the specification of elliptical processes is a core contribution of the paper itself, or whether the focus is on the variational approximation. The current placement of the introduction to $\mathcal{EP}$s in the Background section makes this unclear.

- *[major]* Expanding the Related Work section to more clearly express connections to alternative methods such as deep GPs and deep kernel learning methods.

- *[minor]* Slight tweaks to Experiments section to make key takeaways more clear.

- *[minor]* Tiny spotted typos:

    - pg.1 - "Gaussian distributions *is* attractive..."
    - pg.12 - "a *ten*-fold cross-validation where *we*..."

**Strengths And Weaknesses:**

**Strengths**
- The paper is well-written and a pleasure to read. I believe it does an especially great job in communicating the theoretical contributions required to formulate $\mathcal{EP}$s, while also including multiple illustrative examples that convey the utility of the proposed model in practical set-ups. Crucially, the paper comes across as fairly "complete", and is especially enhanced by the ablation studies and discussion contained in Section 4. I also appreciated that the experiments highlight both strengths and weaknesses of the proposed model (such as the greater risk of overfitting when accounting for heteroscedastic noise).

- The problem itself is very interesting and relevant, and as the authors indicate, highly-relevant to practitioners working with data having heavy-tailed distributions (without compromising on the effectiveness of GPs for normal-distributed data). There would definitely be large value in having the model proposed here being integrated in a more generic toolkit such as $sklearn$ in order to improve usability.

**Weaknesses**

- One aspect that sometimes comes across as unclear is the core contribution of itself, i.e. is it the fundamental formulation of elliptical processes, or the incorporation of variational inference to enable tractability? The initial description of $\mathcal{EP}s$ is provided in the *Background* section, but there is no reference to the unpublished arXiv manuscript on a similar topic (https://arxiv.org/abs/2003.07201).

- A small suggestion for the Experiments section is to perhaps include a critical difference diagram that can more succinctly summarise how GPs and different variations of $\mathcal{EP}s$ compare against each other. Beyond commenting on computational considerations, a more formal description of computational complexity would also be welcome.

- In the related work section, although there are several references to alternative models permitting more complex and flexible posteriors than standard GPs (such as deep GPs), these methods are not featured in the experimental evaluation itself. While I wouldn't expect the evaluation section to be extended further, a little more commentary on what performance can be expected form these competing models when applied to heavily-tailed distributions could be just as illuminating.

- [nitpick] The final sentence of the paper reads as unnecessarily pessimistic, and could with either a citation or a supporting argument.

---

> ### Author Response · Authors · 2023-05-18
> **Reponse to review**
>
> Thank you for reviewing our manuscript. We appreciate your positive feedback and are pleased to hear that you found the paper well-written and enjoyable to read. We have carefully considered your concerns and have made adjustments to address them. Please refer to the updated version of the paper to see all changes.
>
> > Clarifying whether the specification of elliptical processes is a core contribution of the paper itself.
>
> The elliptical process is known in the statistics literature, but there it hasn’t been trained on data the way we do in machine learning. To allow this we introduce the parameterization of the mixing distribution using a normalizing flow. The variational approach is then necessary to perform the training.
>
> >Expanding the Related Work section to more clearly express connections to alternative methods such as deep GPs and deep kernel learning methods.
>
> We have extended the related work section to include various deep alternatives.
> However, without extensive experimentation, we find it hard to speculate on how such deep alternatives would behave when applied to data with heavy-tailed noise.
>
> Thanks again for the comments!

---

### Review · Reviewer_j7Xo · 2023-04-23

**Summary Of Contributions:**

The paper proposes the use of the elliptical distribution for a priori modeling and a posteriori approximation of random processes and observations. Noting that a (consistent) elliptical distribution only arises as a scale mixture of Gaussians, the paper proposes modeling elliptical distributions generically using a normalizing flow for mixing distribution (for both a priori and a posteriori cases). A sparse variational solution for the case of elliptical process prior is provided. Synthetic experiments in regression exploring elliptical noise and heteroscedastic noise settings is provided as well as experiments on real regression and classification datasets.

**Audience:**

Yes

**Claims And Evidence:**

No

**Requested Changes:**

Section 2.2, "Prediction" paragraph : Please correct the statement "The conditional distribution can be..."

Section 3 : Please correct the likelihood definitions in (5), (6), (8), and (9) to include an $i$ subscript on the mixing variable or clarify how the mixing distribution can be estimated as in the experiments of Figure 4 from a single draw of the mixing distribution for all observations. In other words, I believe the correct expression for the likelihood term in (5) is $\prod_{i=1}^N p(y_i | f_i, \omega_i) p(\omega_i; \eta_\omega)$.

Section 3.2 : Please provide some motivatation for selecting a spline flow for the variational posterior $q(\xi; \phi_\xi)$. Since the prior model (5) selects a single draw from the mixing distribution $p(\xi; \eta_\xi)$ it isn't clear why a complicated posterior is necessary.

Section 4.4 : If the paper would like to include a comparison among sparse variants, please select a dataset large enough to warrant them, e.g., the airlines dataset of the sparse GP classification paper by Hensman (2015) et al.

Section 4.4 : Please comment on the estimated test-set log predictive densities of the two (full) EP methods compared to the GP method (or include results and discussion).

Appendix D : Please explain the distinction b/w $\eta$ in (48) and $\eta_f$ in (49).

Appendix G : Please elaborate on the choices of output transformations utilized in the likelihood mixing distribution (softplus) and posterior mixing distribution (sigmoid). The included hypothesis that the posterior is inherently "more difficult to learn" should be backed up with experimental evidence.

Minor:

Section 3 : "...is a regular EP prior with the covariance kernel..." -> a regular GP prior (?)

Section 3.2 : Please modify the title "Prior" of this section. It is not obvious why this title applies.

Section 3.2 : Extra vertical bar in (17) in expression $p(u, \xi |; \eta_u, \eta_\xi)$.

Section 4.2, 1st para : The expression for the true function $f( x_n )$ is missing the additive noise.

Section 4.4, 2nd para : "Therefore, we can indicate the value of the elliptical..." What does it mean to "indicate the value" of the process? Please clarify or re-word.

Section 4.4, 3rd para : "performed a ton-fold cross-validation" -> ten-fold

Appendix B : At the end of the line starting with "Proof of proposition 1.", the variable $M$ in $p(\xi|y_1 M ; \eta)$ is undefined.

Appendix F : There is a missing $\xi$ in the expression in the line above (66).

**Strengths And Weaknesses:**

Strengths

Synthetic evidence :

I liked the experiment of Figure 5 where the consequence of utilizing the more straightforward GP w/ Gaussian likelihood results in an overfit latent function with smaller estimated lengthscales. This is among the clearest examples provided for how using an elliptical distribution for noise can benefit modeling. The heteroscedastic experiment in Section 4.2 is also nicely illustrative.


Weaknesses

Clarity :

For the use case where an elliptical distribution is used as observation likelihood, there is either reader confusion in interpreting the joint probability (5) and noise likelihood (6) or typos in these equations : The mixing distribution $p(\omega; \eta_\omega)$ for the likelihood can be defined either per observation or hierarchically as (5) and (6) specify. In the former case, the mixing distribution can be estimated from observations IF the same elliptical distribution is used across observations and IF it happens to be true that the equivalence of two univariate elliptical distributions implies equivalence of their mixing distributions. In the latter (hierarchical) case, the mixing distribution cannot be estimated from observations for the same reason that it cannot be estimated when an elliptical distribution is used to model a random process, i.e., only a single draw of the mixing variable is observed. Figure 3a,c and the experiments of Figure 4 imply the paper is assuming the former model, but the a priori and a posteriori model specifications are for the latter model.

For the use case where an elliptical distribution is used within a random process prior, there are some inconsistencies in the exposition that create confusion. In Section 2.2, "Prediction" paragraph, in the statement "The conditional distribution can be derived analytically (see Appendix B), but we will instead solve it by approximating the posterior $p(\xi|y; \eta_\xi)" what does "conditional distribution" refer to? The results of Appendix B refer to a pure elliptical distribution while the statement of Section 2.2 refers to conditioning on observations, i.e., drawn from a pure elliptical distribution plus i.i.d. noise.

Experimental methodology :

The paper does not provide a procedure for how the number of bins in an a priori or a posteriori mixing distribution spine flow should be set and how an output transformation (softplus, sigmoid, arctan) after the flow should be specified. In other words, it should be clear to a reader whether or not the settings described in appendix G were tuned for the experiments (preferably, not tuned). Similarly, for implementing a heteroscedastic elliptical distribution noise model, how would one select an architecture for the input-varying network model $g_{\gamma_\omega}( )$ of Section 3.3 in a novel problem?

Regression experiments :

The EP-EP results w.r.t. MSE plotted in Figure 7 are worse than the exact GP results in 5/7 datasets and not very different in the remaining 2. The claim that "...log-likelihood is more critical than MSE because mean-squared error (MSE) does not consider the predictive variance" is not unconditionally true. One would have to show that, for a given value of MSE, no value of variance could yield a given log likelihood. In other words, both metrics generally provide information. In the 5/7 cases where the exact GP has worse log likelihood, it could be argued that the exact GP is somehow poorly calibrated. The EP-GP method does not seem to perform any better than the EP-EP method across the metrics and datasets. The heteroscedastic EP method appears to be trading log likelihood for MSE even more than the vanilla EP-EP method. Finally, it isn't clear what additional value SVGP provides for interpreting the proposed method's performance.

Classification experiments :

The evidence among the non-sparse variants in support of the proposed method seems marginal. One dataset (mammographic) shows little difference among the methods in both performance metrics. Another dataset (heart) shows some improvement in one metric (accuracy) but not much in the other (AUC). The final dataset (pima) does show what appears to be statistically significant improvement in one metric (AUC). In total, one could say that the use of EP distributions does not hurt modeling binary classification and can sometimes provide a little benefit. The included sparse variants are not necessary since the datasets are all small (<1000 sample size) and obfuscate interpreting the results.

---

> ### Author Response · Authors · 2023-05-18
> **Reponse to review**
>
> Thank you for taking the time to read our paper so thoroughly and for pointing out some of the shortcomings of the paper. We have made a number of changes which we hope clear up some of the confusion. Please refer to the updated version of the paper to see all changes.
>
> >Section 2.2, "Prediction" paragraph: Please correct the statement "The conditional distribution can be..
>
> Thank you for spotting this. The conditional distribution in Appendix B does not include the additive noise but refers to noise-free data. It could also be used if we add the noise to the kernel, as they do in the Student-t process paper. We have included the following comment in the paper to clarify this:
>
> "The noise-free predictive distribution can be derived analytically (see Appendix B), but to be able to account for additive elliptical noise, we will instead solve it by approximating the posterior $p(\xi| \, y ; \eta_\xi)$ with a variational distribution $q(\xi ; \varphi_\xi)$."
>
>
> >Please correct the likelihood definitions in (5), (6), (8), and (9) to include a subscript on the mixing variable.
>
>
> You are correct in your interpretation that the omega is sampled per observation. To clarify this, we have added subscripts to omega as suggested.
>
> >The paper does not provide a procedure for how the number of bins in an a priori or a posteriori mixing distribution spine flow should be set
>
> These parameters were not tuned but fixed for all experiments. We have added a sentence to clarify this in the appendix.
>
> >Section 3.2: Please provide some motivation for selecting a spline flow for the variational posterior.
>
> If a certain model leads to a known expression for the posterior, that could of course be used instead of the spline flow. However, in general, we only know that the posterior is elliptical with a data-dependent mixing distribution. Since it is impossible to “overfit” a variational distribution (Bishop p. 464) we choose a very flexible yet tractable model, namely a spline flow.
>
>
> >In the classification results, If the paper would like to include a comparison among sparse variants, please select a dataset large enough to warrant them and add and comment on the estimated test-set log predictive densities.
>
> Thanks for your suggestions. We removed the sparse versions of the models in the plots and added a plot of the test log-likelihood.
>
> >Please explain the distinction b/w in (48) and in (49).
>
> We rewrote the equations to make the distinction more clear.
>
>
> >Appendix G: Please elaborate on the choices of output transformations utilized in the likelihood mixing distribution (softplus) and posterior mixing distribution (sigmoid).
>
> The output transformations were set early on in the process and haven't changed since then. Although testing different output transformations of the mixing distribution would be interesting to investigate, we consider it outside the scope of this paper. Given that this is merely a design choice and not a hypothesis we test, we removed the sentence “The reason we use a \emph{Sigmoid} and only five bins for the posterior is that we want to regularize it more since we hypothesize it is more difficult to learn”.
>
>
> Again, thank you for your comments!

---

### Review · Reviewer_bt3E · 2023-05-05

**Summary Of Contributions:**

The paper proposes elliptical processes (EPs), a new family of stochastic processes inspired on GPs (Rasmussen & Williams, 2006) and Student-T processes (SPs) (Shah, ICML 2014). The underlying principle of these new EPs is that they are based on elliptical distributions, which are a more general family of distributions, typically more heavy-tailed, that includes Student-T, Cauchy as well as the Gaussian density. The key point of the elliptical distributions is their density generator, which in the end, depends of a parametric mixing variable \xi. The authors consider this mechanism for both the function prior and the likelihood distribution, as they hope to better capture the uncertainty with the parametric heavy-tailed distribution. For the mixing distribution on the likelihood function, they consider a normalising flow based on splines. Experimental results show some evidence of the performance in regression, classification and heteroscedastic tasks, mainly in the well-known UCI datasets.

**Audience:**

Yes

**Broader Impact Concerns:**

I do not detect particular ethical implications of the work.

**Claims And Evidence:**

Yes

**Requested Changes:**

Authors followed the requested changes {1,2} in the previous revision (see my previous comments as Reviewer CEkx), which have now improved quite a lot the quality of the manuscript. Additionally, removing the multi-task details made the manuscript much clearer and detailed, allowing the reader to focus on the main contributions and the technical utility of the proposed method.

To me, even if some extra experiments have been added, I still have a general concern around the limitation of the empirical results. To give an example, the performance of HetEP, EP-EP and EP-GP is kind of in-between the rest of SOTA metrics in Figure 7. Even if some analysis/conclusions are included in the end of Section 4.3, this is not entirely sufficient to me. I remark that I am not looking for metrics or datasets that indicate a better performance for the proposed model, but I would like to see that there is a clear utility on using the model indicated by some empirical results.

> One advise would be to think about scenarios, datasets or problems (i.e. regression) where the use of Gaussians is predominantly worse than other heavy-tailed distributions. The authors indicated "finance" as one of the applications for more heavy-tailed stochastic processes, perhaps that could be a good one. But at least, I would like to see a real example of this effect and how the proposed EP correctly captures the whole uncertainty, improving standard GPs.

> Other extra (minor) points could be to improve the explanations around the effect of using the elliptical distribution in the likelihood and the function prior, for example. Also the presentation of the sparse/inducing points could be fixed a bit, particularly the use of X_u and x_n, which seemed a bit odd to me.



**Strengths And Weaknesses:**

The paper brings an interesting topic to the audience of GP models, which is the problem of correctly calibrating the uncertainty in probabilistic non-linear regression and classification tasks. The use of more heavy-tailed distributions instead of just Gaussian densities makes sense to me, and authors motivated it better in this new version of the manuscript. The general presentation of the method and the technical details are clear, and easy to follow, apart from some details around the concept of identifiability and the theoretical conditions to be a stochastic process in Section 2.2, which are a bit short and unclear to me. (More details around these two points would be highly appreciated). On the modelling side, I saw the main point of the model, playing around with the parametric mixing distribution to control the elliptical model in both the likelihood and prior functions of the GP (or EP). The application of the proposed model to several datasets and comparison with SOTA methods is also a valuable point that I remark.

---

> ### Author Response · Authors · 2023-05-18
> **Reponse to review**
>
> Thanks for, again reading the paper so thoroughly and giving us fruitful feedback on the paper. We are glad to hear that you think technical details are clear and easy to follow. We have uploaded a new version of the paper. Below we comment on your main concerns:
>
>
> >More details around the concept of identifiability and the theoretical conditions to be a stochastic process in Section 2.2.
>
> Response: We rewrote the identifiability paragraph to, hopefully, make it more clear.
>
> >Wants to see a clear utility in using the model indicated by some empirical results.
>
> To illustrate the potential usefulness of the elliptical process we added a new section (4.4) to the experiments where we consider probabilistic forecasting of wind power production. In this scenario, the full distributions are important for improving the reliability of production planning. We show that the EP-EP model works well, and include comparisons in the appendix showing that Het-EP performs similarly while the exact GP is underconfident and overfits to short-scale trends.
>
> >The use of X_u and x_n, as a bit odd
>
> Response: Thank you for the comment. We replaced X_u with Z (the induced locations) and x_n to x_i.
>
> Thanks again for the comments which we feel improved the manuscript!

---

### Public Comment · ~Laurence_Aitchison1 · 2023-03-04
**Suggested reference.**

Might be worth citing deep kernel processes, as they are also non-parametric probabilistic models that subsume the Gaussian processes and the Student's t processes (arxiv.org/abs/2010.01590).

---

### Decision · Action_Editors · 2023-06-28

**Recommendation:** Accept with minor revision

**Comment:**

Overall, this paper is of interest to the community and the clarity and focus of the paper are much improved from the previous submission. Indeed, those reviewers who reviewed both versions were very complementary about the improvements in this version.

While there are some concerns about the experimental results (which show marginal improvements over alternative methods), the reviewers all agree that EPs, and the proposed methodology, are of interest to the community. Therefore, I do not request any further experiments.

I *would* like to see clearer delineation of the contributions. Two out of three reviewers were of the impression that the Elliptical process was a new concept; however as the authors address in response to reviewer g9nQ, the idea exists already in the statistical literature (albeit, not as a practical modeling method). Making EPs a practical modeling tool is certainly an important contribution, and under "Our Contributions", the authors do indeed say that their contribution is a new construction of EPs, not the introduction of EPs. However, I would like to see this distinction made clearer in the Introduction and in Section 2.2 (where it is unclear whether Definition 1 is a new contributions since it has no attribution).

In addition, the reviewers noted that, while the paper is much approved, it would benefit from a final proof read -- for example, in (5) the subscript should be $n$ not $i$

**Audience:**

Yes, this paper should be of interest to researchers working on Gaussian processes and their extensions.

**Claims And Evidence:**

The authors present a new construction and variational inference algorithm for elliptical processes (EPs), a general class of heavy-tailed stochastic processes that includes Gaussian and Student-t processes. They show that EPs can be formulated as a scale mixture of Gaussian processes, and use a normalizing flow to model the mixing distribution.

The authors do a good job of motivating EPs, and provide clear illustrative examples. In general, the motivation is much clearer than in the authors' previous submission. The real-world experiments do not fully live up to the promise of the theory and synthetic experiments, with the practical benefit over alternative methods apparently small.

Despite this, the reviewers and I feel this paper is of interest to the community: it introduces a new approach for function modeling, and provides a well thought out inference algorithm. It is likely that this work will spur further research into Elliptical Processes.